# Molecular elucidation of drug-induced abnormal assemblies of the hepatitis B virus capsid protein by solid-state NMR

Lauriane Lecoq [1], Louis Brigandat[1], Rebecca Huber[1], Marie-Laure Fogeron[1], Shishan Wang[1], Marie Dujardin[1], Mathilde Briday [1], Thomas Wiegand[2,7,8], Morgane Callon[2], Alexander Malär[2], David Durantel [3], Dara Burdette[4], Jan Martin Berke[5], Beat H. Meier [2] ✉, Michael Nassal [6] ✉ & Anja Böckmann [1] ✉

Hepatitis B virus (HBV) capsid assembly modulators (CAMs) represent a recent class of anti-HBV antivirals. CAMs disturb proper nucleocapsid assembly, by inducing formation of either aberrant assemblies (CAM-A) or of apparently normal but genome-less empty capsids (CAM-E). Classical structural approaches have revealed the CAM binding sites on the capsid protein (Cp), but conformational information on the CAM-induced off-path aberrant assemblies is lacking. Here we show that solid-state NMR can provide such information, including for wild-type full-length Cp183, and we find that in these assemblies, the asymmetric unit comprises a single Cp molecule rather than the four quasi-equivalent conformers typical for the icosahedral T = 4 symmetry of the normal HBV capsids. Furthermore, while in contrast to truncated Cp149, full-length Cp183 assemblies appear, on the mesoscopic level, unaffected by CAM-A, NMR reveals that on the molecular level, Cp183 assemblies are equally aberrant. Finally, we use a eukaryotic cell-free system to reveal how CAMs modulate capsid-RNA interactions and capsid phosphorylation. Our results establish a structural view on assembly modulation of the HBV capsid, and they provide a rationale for recently observed differences between in-cell versus in vitro capsid assembly modulation.

Chronic infection with hepatitis B virus (HBV), a small enveloped retrotranscribing DNA virus, afflicts more than 250 million people. With nearly one million deaths per year, the resulting chronic hepatitis B (CHB) represents one of the most deadly diseases, especially in Africa and the Western Pacific region[1]. While an effective prophylactic vaccine is available, current therapies can rarely cure infection. Owing to severe side-effects, only few patients are eligible for the finite treatment with type-I interferon (IFN), whereas nucleos(t)ide analogs (NAs) inhibiting the viral polymerase usually require life-long administration to keep virus replication suppressed; otherwise, the covalently closed circular (ccc) DNA, the persistent nuclear form of the viral genome, will resume to give rise to progeny virions[2]. Hence multiple efforts are

[1]Molecular Microbiology and Structural Biochemistry (MMSB), Labex Ecofect, UMR 5086 CNRS/Université de Lyon, 69367 Lyon, France. [2]Physical Chemistry, ETH Zurich, 8093 Zurich, Switzerland. [3]Centre de recherche en cancérologie de Lyon (CRCL), UMR 5286, Centre Léon Bérard, 69373 Lyon, France. [4]Gilead Sciences, Foster, CA, USA. [5]Janssen Pharmaceutica N.V, Beerse, Belgium. [6]Dept. of Medicine II/Molecular Biology, University of Freiburg, Freiburg, Germany. [7]Present address: Max-Planck-Institute for Chemical Energy Conversion, Stiftstr. 34-36, 45470 Mülheim an der Ruhr, Germany. [8]Present address: Institute of Technical and Macromolecular Chemistry, RWTH Aachen University, Worringerweg 2, 52074 Aachen, Germany. ✉e-mail: beme@ethz.ch; michael.nassal@uniklinik-freiburg.de; a.bockmann@ibcp.fr

underway to achieve a functional cure, i.e. maintained viral suppression upon therapy withdrawal, or ideally a sterilizing cure, i.e. elimination of the virus[3]. Besides targeting HBV-relevant host factors[4], direct-acting antivirals are being investigated[5]. One of the most advanced new drug classes are capsid assembly modulators (CAMs)[6,7], sometimes called core protein allosteric modulators (CpAMs). They target the core protein (Cp), the building block of the icosahedral HBV capsid, whose pleiotropic activities are crucial for multiple steps of the viral life cycle[8,9], including as a specialized compartment for genome replication[7]. Cp comprises an N-terminal assembly domain (residues 1–140) connected by a nine-residue linker to a highly basic C-terminal domain (CTD; residues 150–183) which binds nucleic acids, notably the viral pregenomic (pg) RNA[10], the substrate for capsid-internal reverse transcription into the HBV-typical partially double-stranded (ds) relaxed circular (rc) DNA in infectious virions[11]. Cp's assembly domain features five α-helices, as determined by early cryo-EM and x-ray studies[12,13] of recombinant capsids from truncated CTD-less Cp. The central helices α3 and α4 from two Cp molecules associate into a four-helix bundle, forming a stable dimer; these associations form the spikes which are a prominent feature of the capsid shell. The Cp dimers then assemble into closed icosahedral shells. The major class comprises 120 Cp dimers arranged with triangulation number $T = 4$ symmetry, a minor class ($T = 3$) consists of only 90 dimers. In the simplest case of icosahedral symmetry, $T = 1$, three structurally identical capsid protein subunits contribute each of the 20 triangular faces of the icosahedron. Larger shells accommodating more than 60 subunits are accessible based on the principle of quasi-equivalence[14], whereby each face is further subdivided by a certain integer. The respective triangulation ($T$) number represents the number of similarly but non-identically structured ("quasi-equivalent") subunits required to adapt to the same number of symmetry-related non-identical environments. Hence for $T = 4$ symmetry there are four different molecules or, for viruses with a single capsid protein like HBV, four conformers in the asymmetric unit, termed A, B, C and D, while artificial sheet-like structures can be obtained in crystal showing also E and F subunits (Fig. 1a, b).

Cp plays important roles in almost every step of the HBV life cycle, including pgRNA co-encapsidation with the viral polymerase, enabling capsid-internal pgRNA reverse transcription into relaxed circular (rc)DNA, signaling for capsid envelopment with the surface proteins, and virion secretion. Typically, the Cp spikes have recently been described to be involved in nuclear export of the encapsidated viral RNA, with the tip containing a sensor for secretion[15,16]. While the flexibility of the spike tips has been repeatedly reported[17–19], its functional importance is not firmly settled, but might be important in adequately presenting the nuclear export signals, and also other envelopment-relevant motifs. Also, the capsid carries a continuous flow of rcDNA into the nucleus via the core protein's nuclear localization signals located in the CTD. With few recent exceptions[20,21] most structural studies were done on CTD-less Cp variants, hence the underlying structural dynamics are not yet understood in detail, but they must be tightly regulated. Interfering with proper nucleocapsid assembly and its regulation is therefore a promising antiviral strategy (see ref. [7] for a recent review).

The first small molecule capsid assembly modulators (CAMs) were phenylpropenamide derivatives such as AT-130[22] which promote formation of pgRNA-less empty capsids, and hetero-aryldihydropyrimidines (HAPs), e.g. BAY 41-4109[23] which can induce formation of aberrant, nonspherical multimers which are neither functional in virus replication[24,25]. These in vitro phenotypes, though exerting some concentration dependence[26], are prototypical for the two main categories of CAMs[27]: on one hand, CAM-E compounds, such as phenylpropenamides and sulfamoylbenzamides including JNJ-632 and JNJ-827, that cause assembly of empty capsids of apparently normal shape; on the other hand, CAM-A compounds such as

HAPs, including JNJ-890, that lead to aberrant assemblies[28–31]. An earlier classification into class I and class II CAMs was applied in a non-uniform manner and led to confusion in the past[32].

As neither empty nor aberrant nonspherical particles are compatible with containment of the pgRNA−viral polymerase complex, both CAM classes prevent the formation of rcDNA. Moreover, CAMs have been proposed to decrease the formation of cccDNA and to restore innate signaling, which is not achieved by current NA treatments[28]. CAMs are thus considered as some of the most potent candidates for a curative hepatitis B treatment[6,7], and several compounds are in clinical trials[32,33].

Structures of capsids in the presence of CAMs have been determined both by X-ray and cryo-EM[26,31,34–38]. Accordingly, despite their different impact on overall particle morphology, both types of CAMs bind to the same hydrophobic cavity at the Cp dimer-dimer interface, called the HAP-binding pocket (Fig. 1a, b). It is described to consist of a concave interface involving residues F23, P25, D29, L30, T33, L37 (comprising mainly helix 2), W102, I105, S106, T109, F110, Y118, F122, I139, L140 and S141 (comprising helices 4 and 5 and linker region) from one chain, and the region from residue V124 to P134 (helix 5 and linker region) from the other chain[6].

Notably, however, as both X-ray crystallography and cryo-EM require regularly ordered specimens, the aberrant Cp assemblies induced by CAM-A compounds are intrinsically unsuited for these high-resolution analyses. Hence, in all these studies, recombinant Cp mutants were used which either enabled crystallization in a non-particulate planar hexagonal form via the capsid assembly-preventing Y132A mutation[31,36,39], or artificially forced to maintain near-icosahedral symmetry through intra-capsid cross linking of extra cysteine residues added as residue 150 to the C terminus truncated Cp149 (Cp150)[26,34,35,37]. In the Y132A mutant, each of the three dimer-dimer interfaces in the hexamer accommodates one CAM (Fig. 1a), while in the Cp150 capsid, only one CAM binds per dimer in the B and C subunits (Fig. 1b); how this relates to CAM binding to wild-type Cp remains to be determined, because as the aberrant, irregular assemblies observed upon addition of CAM-A to wild-type Cp149 and alike have not been investigated by any direct structural approaches. Furthermore, truncated Cps lacking the entire CTD were used in nearly all studies, hence very little structural information is available for the full-length wild-type Cp183 protein, and whether and how CAM binding is affected by the CTD. Solid-state NMR can structurally analyze large protein assemblies without the need for crystals or symmetry, as has been shown for protein fibrils (reviewed e.g. in ref. [40]), membrane proteins (reviewed e.g. in ref. [41]), molecular machines (e.g.[42,43]) and viral capsids and envelopes (reviewed e.g. in[44]). NMR chemical shifts are highly sensitive environmental indicators, and can report on conformational variations and molecular binders[44,45].

In this work, we use this approach to investigate the conformational impact of CAMs on the irregular assemblies which form in their presence, including from full-length wild-type Cp183 with RNA, and also the phosphorylated, nucleic acid-free form P7-Cp183. In addition to incorporation of CAMs into preformed capsids, we also use wheat-germ cell-free protein synthesis (WG-CFPS) to study the effects of CAMs on Cp when present in situ during assembly upon Cp exit from the ribosome. Our study provides a molecular-level observation of CAM assembly modulation for full-length Cp carrying a functional CTD, and shows that the data on the impact of CAM on Cp149 capsids do not tell the whole story.

## Results

### CAM-A induce Cp assemblies with a single conformer

Negative-staining EM of Cp in the presence of CAM-A compounds has revealed irregularly-shaped structures, for example induced by HAP compounds[46,47]. For illustration, Fig. 1c (right panel; as opposed to intact capsids in the left panel) shows such micrographs of Cp149

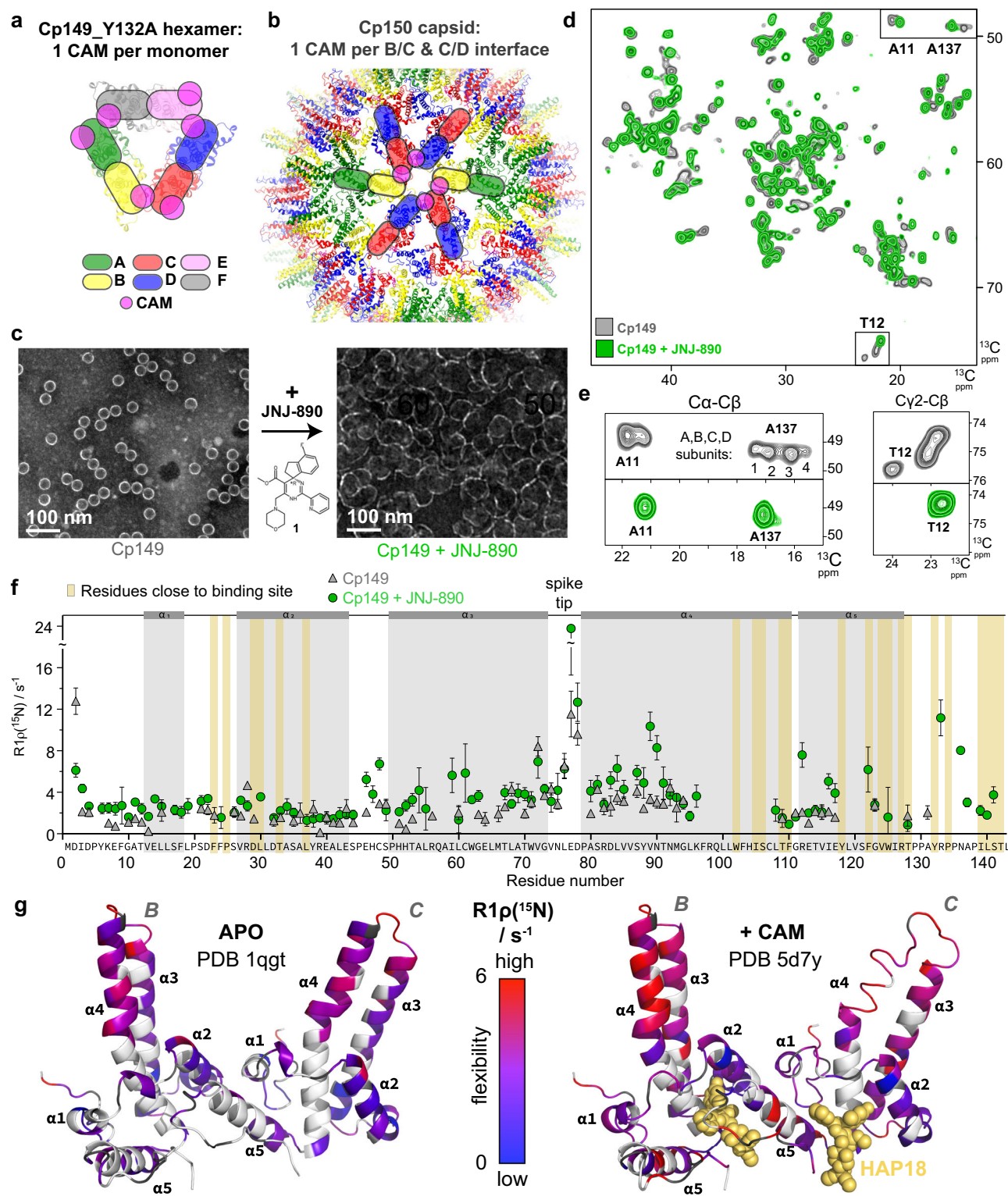

**f** Residues close to binding site — Cp149 (gray triangles); Cp149 + JNJ-890 (green circles)

**g** APO PDB 1qgt — +CAM PDB 5d7y — R1ρ(¹⁵N) / s⁻¹ high (flexibility) to low. HAP18

capsids in presence of JNJ-890, a CAM-A compound (see Table S1 for the list of CAM compounds used and Table S2 for the full list of forms investigated). The mesoscopic heterogeneity of the observed capsids made us suspect that NMR spectra might show severe heterogenous line broadening, representing a distribution of different protein conformations. In order to assess the structural features at a molecular level, we recorded 2D Dipolar Assisted Rotational Resonance (DARR) spectra (Fig. 1d, Supplementary Table 3) and hNH (Supplementary Fig. 1, Table S4) nitrogen-proton correlation spectra of Cp149 capsids in presence of JNJ-890 (in green), and compared them to spectra

without CAM-A (in gray). The spectra surprisingly demonstrate that the aberrant objects induced by the CAM-A compound actually yield well-resolved resonance lines in the solid-state NMR spectra, with lines which are even narrower than in the absence of CAMs. Of note, whether CAMs were added to Cp149 preassembled capsids, or to Cp149 dimers before capsid assembly, made no difference for the resulting spectra (Supplementary Fig. 2), as also observed by EM[48]. Sequential resonance assignments using 3D NMR following previously described protocols[49,50] revealed only a single peak per atom for the CAM-A capsid spectra, while we previously described for the apo form that

**Fig. 1 | Effect of JNJ-890 binding on Cp149 structure and dynamics. a** In the Cp149_Y132A crystalline hexamer, each monomer can accommodate one CAM. In addition to the A, green; B, yellow; C, red; and D, blue subunits of $T = 4$ icosahedral capsids, E, pink; and F, gray; subunits are present in the asymmetric unit of the Y132A hexamer crystal trimer of dimers (e.g. PBD 5wre[31]). **b** Representation of the localization of CAM binding sites (pink circles) found in the A, green; B, yellow; C, red; and D, blue; subunits of HBV Cp150 capsid stabilized by artificial disulfide bridges involving C150 (e.g. PBD 5d7y[37]). **c** Negative-staining EM micrographs of Cp149 NMR samples in the absence and presence of the JNJ-890 CAM-A. The experiment was repeated independently at least 10 times with similar results. **d** Overlay of aliphatic spectral regions of $^{13}C$-$^{13}C$ DARR spectra of Cp149 in absence (gray, from ref. [53]) and in presence of JNJ-890 (green). **e** Extracts of the peak multiples corresponding to the different conformers in the asymmetric unit of T12 and A137 resonances, and the resulting single resonance in presence of CAM-A (from

ref. [53]). **f** Rotating-frame relaxation-rate constants $R_{1\rho}(^{15}N)$ in the absence (gray triangles) and presence (green circles) of JNJ-890 (for relaxation-rate-constant differences see Supplementary Fig. 3, and experimental details in Table S5). $n = 1$ independent experiments have been recorded. The dots and triangles represent fitted relaxation-rate constants $R_{1\rho}(^{15}N) \pm 2$ times the standard deviation (see material and "Methods" for details on $R_{1\rho}(^{15}N)$ measurements and Supplementary Fig. 3c, d for individual fits). The error bars are derived from a bootstrap procedure. Source data are provided in the Source Data file. The larger the rate constant, the larger the molecular dynamics on the microsecond time scale. **g** Experimentally elucidated dynamics plotted on the apo-capsid structure (PDB 1qgt[13], left structure) and on a CAM-A-bound structure (PDB 5d7y[37] with HAP18 in yellow spheres, right structure), where red indicates high amplitude motions on the microsecond time scale and blue low amplitude motions. Prolines are colored in gray and residues with unavailable data in white.

many amino-acid residues displayed resolved peaks for the four different molecules in the asymmetric unit, A, B, C and D of the $T = 4$ icosahedral capsid[51]. We have shown that the resulting peak splitting is strongest for amino-acid residues located at the dimer-dimer interfaces[51], where pentamers and hexamers have in total four slightly distinguishable monomer conformations[13]. The splitting becomes smaller than the linewidth for residues remote from the dimer-dimer interfaces, as for example the amino acids in the spike, resulting only in a certain line broadening. This pattern is lost in CAM-A capsids, as illustrated in the 2D NMR spectra of Fig. 1e for the example of T12 and A137 amino-acid residues, both located at the dimer-dimer interface. Quite unexpectedly, the CAM-A Cp spectra are thus simplified with respect to the apo Cp, and the peaks are particularly narrow. This behavior was observed for all 28 residues which showed peak splitting in the apo form. The flattening out observed in the EM micrographs of the assemblies suggests that the lattice is closer to hexagonal, since such types of lattices form rather flat superstructures, whereas the pentameric axes in icosahedral structures introduce curvature. That the observed objects are not fully flat might indicate that a few pentamers are still present, but at an abundance of below approximately 5% they would be below the NMR detection limit of our NMR experiments. This implies that in the Cp149-JNJ-890 complex, the remaining pentameric structures are present at most at this frequency.

In order to assess the impact of the CAM-A on capsid dynamics, we measured NMR relaxation-rate constants $R_{1\rho}(^{15}N)$ of Cp149 in the presence and absence of JNJ-890. These constants report on the internal flexibility of the NH vector of each amino-acid residue on the microsecond time scale. Overall it appears that the relaxation-rate constants, i.e. the flexibility on the microsecond time scale, increase upon drug binding in most parts of the protein, as shown in Fig. 1f (for a plot of the rate-constant differences see Supplementary Fig. 3a, b, and site-specific fits in Supplementary Fig. 3c, d). In particular, the capsid spike acquires significant mobility in presence of JNJ-890, as shown on the Cp structures in Fig. 1g. In addition, smaller effects are detected in the spike base. Some regions remain mostly unaffected in terms of dynamics, especially helix α2, as well as the end of helix α4′ and the beginning of helix α5. Interestingly, residues close to the flexible linker and the CAM binding site (region 130-142) became more intense upon addition of JNJ-890, possibly through collapse of the split peaks, or following a stabilization upon interaction, enabling the measurement of more data points in this region.

In summary, we surprisingly observe, for the abnormal assemblies resulting from addition of a CAM-A compound, very high-quality NMR spectra. They interestingly reveal a collapse of the formerly split peaks in $T = 4$ icosahedral Cp into single peaks, and also show increased dynamic behavior, notably in the spike region.

## NMR fingerprints directly pinpoint CAM-proximal residues
While negative-staining EM can distinguish between aberrant Cp assemblies formed in the presence of CAMs and normal capsids,

NMR spectra allow to obtain a detailed fingerprint of the interacting residues, and to detect changes in local symmetry, including a different asymmetric unit. Figure 2 shows EM micrographs and sections of NMR spectra (for full aliphatic regions and EM micrographs of sedimented NMR samples see Supplementary Fig. 4a, b) for Cp149 with a variety of CAMs for which the chemical structures are given. While in *apo* Cp149 (top of Fig. 2) capsids, four different A137 and two different A11 peaks are clearly distinguished, the number of peaks decreases when CAMs are added. For GS-942049, GS-832471, and JNJ-827, it can be seen that two out of the previously four peaks disappear. For JNJ-632 and HAP_R10[52], the two peaks start to coalesce. This points to the dimers, which start out as parts of the hexameric and pentameric units in the context of icosahedral symmetry, becoming more similar to each other. Finally, for JNJ-890, only a single major peak is left, indicating a virtually single conformation. It should be noted that the single peak for the JNJ-890 Cp finally indicates full occupancy with CAM on all sites.

Figure 2c shows the residue-specific chemical-shift differences between the *apo* and CAM-bound capsids for the different compounds plotted on the Cp dimer 3D structure (PDB 5d7y[37]) (for chemical-shift perturbations (CSPs) see Supplementary Fig. 5a). CSPs allow to directly identify the residues which are most impacted by the interaction with a CAM. As an example, a zoom on the peak in 2D NCA spectra of Y132, a residue located at the inter-dimer interface, is shown in Supplementary Fig. 5c for several Cp samples in presence of CAM-A. Overall, a number of similar CSPs is observed for both CAM-A and CAM-E, highlighted in yellow in Supplementary Fig. 5a, b. Interestingly, F24, L30, P129 and Y132 show significantly larger CSPs for CAM-A than for CAM-E (green in Supplementary Fig. 5a, b), and N136 and P138 show large CSPs only in presence of CAM-E (pink in Supplementary Fig. 5a, b). Large CSPs in these resonances can thus be considered as discriminators for CAM-E *vs* CAM-A compounds. Interactions between the CAMs and these residues might be responsible for the divergent action of the two classes of molecules. The largest number of significant CSPs is observed for the CAM-A compounds, likely since CSPs not only reflect the binding of CAMs, but also changes in symmetry. Notably, the CSPs observed for CAMs differ from those observed for Triton-X100[53], which binds to the hydrophobic pocket located at the intra-dimer interface, whereas CAMs bind to the hydrophobic cavity located at the inter-dimer interface.

To summarize, NMR can distinguish different classes of CAMs by their spectral fingerprint: CAM-E show distinct patterns of multiple peaks, while the aberrant assemblies resulting from CAM-A show a major single resonance per amino acid, and the largest CSPs, also affecting the largest number of residues. This atomic-level fingerprint presents an interesting complement to kinetic methods (e.g. microfluidic screening[54]) used to distinguish CAM-A from CAM-E compounds.

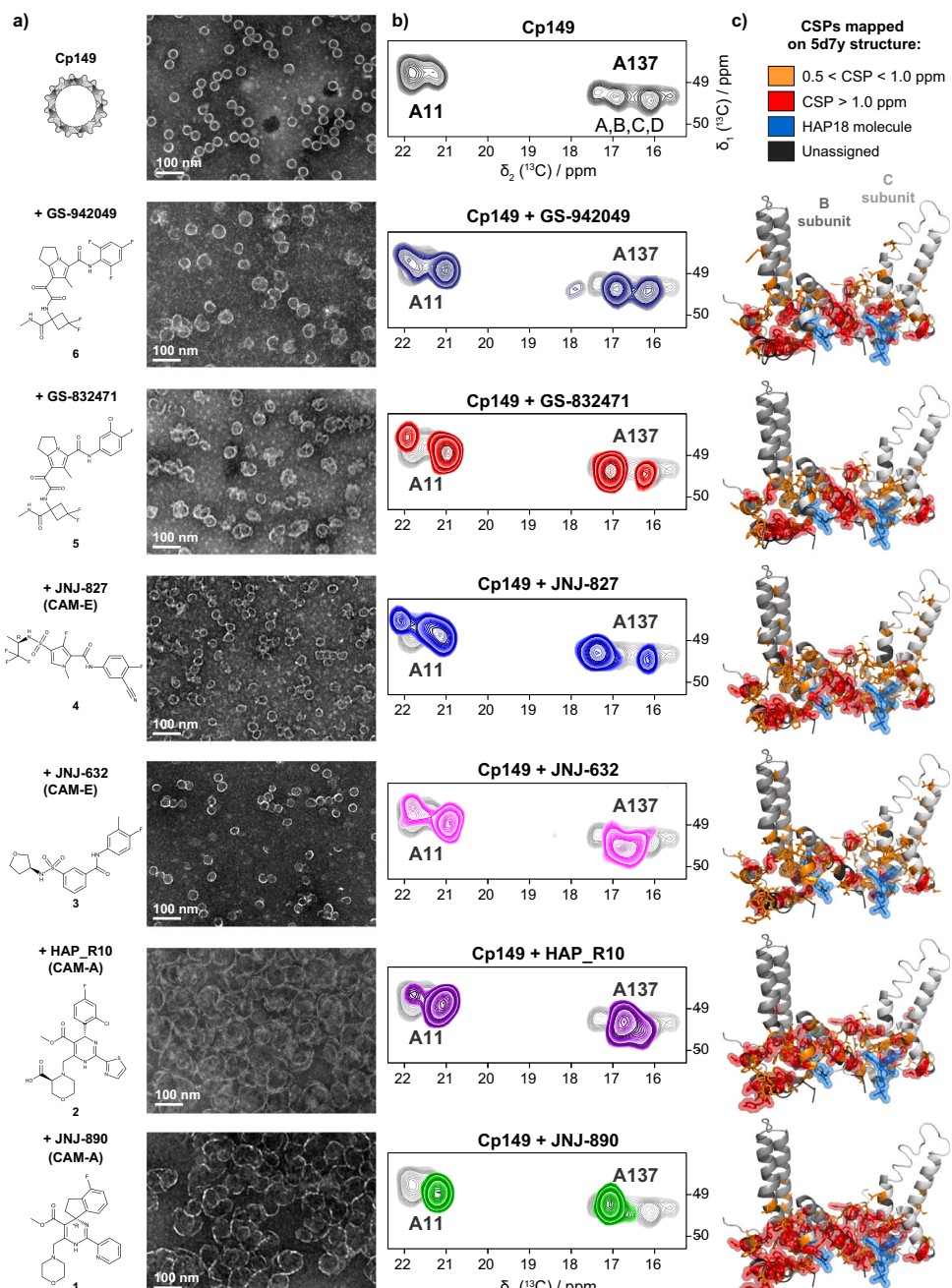

**Fig. 2 | CSPs induced by different CAM-A and CAM-E compounds.** a Negative-staining EM micrographs corresponding to the $^{13}$C-$^{15}$N Cp149 reassembled capsids in absence (for reference) or presence of CAM, with their chemical structure. The EM pictures were taken on single labeled samples, and experiments were repeated independently at least once on unlabeled samples and yielded similar results. **b** Extracts from the NMR DARR spectra showing the A11 and A137 alanine Cα-Cβ region for all tested compounds. In light gray is the control spectrum for

comparison (from ref. [53]). For full aliphatic regions of DARR spectra, see Supplementary Fig. 4a. **c** CSPs induced by the different CAMs mapped on Cp149-HAP18 structure (PDB 5d7y[37]) where unaffected residues are shown in gray, and residues affected by the binding are colored orange (medium CSPs, in sticks) and red (strong CSPs, in spheres). HAP18 is represented as blue spheres. For CSP graphs for all residues see Supplementary Fig. 5 and values in the Source Data file.

## CAM-A bound Cp183 and Cp149 are different at a mesoscopic but identical at the molecular level

A question not answered before is how CAMs affect, at the molecular level, the structure of capsids assembled from the full-length Cp183 protein. We thus analyzed, in addition to truncated Cp140 and Cp149, Cp183 and phosphorylated P7-Cp183, prepared by sedimentation[55,56] of the capsids into the NMR rotor in the presence of JNJ-890 CAM-A. Figure 3a shows the EM micrographs of the different capsids after incubation with JNJ-890. The superstructures formed are for Cp140 and Cp149 clearly open, while full-length capsids seem nearly unaffected,

also at higher Cp:CAM-A ratios (Supplementary Fig. 6). Notably, P7-Cp183 capsids (Supplementary Fig. 6c) remained mainly circular, even if some seem to have disrupted contours. Even at very high ratios, up to 1:20, their diameters seem to increase slightly, but they do not substantially open (Supplementary Fig. 7). In contrast, Cp149 already at a ratio Cp:JNJ-890 of 1:1 has nearly disassembled (Supplementary Fig. 6b). When analyzing changes at different points in time (Supplementary Fig. 8), Cp183 capsids started to detectably open up after one month of storage, but still showed a more compact structure than Cp149 with CAM-A. Altogether this indicates that there are additional

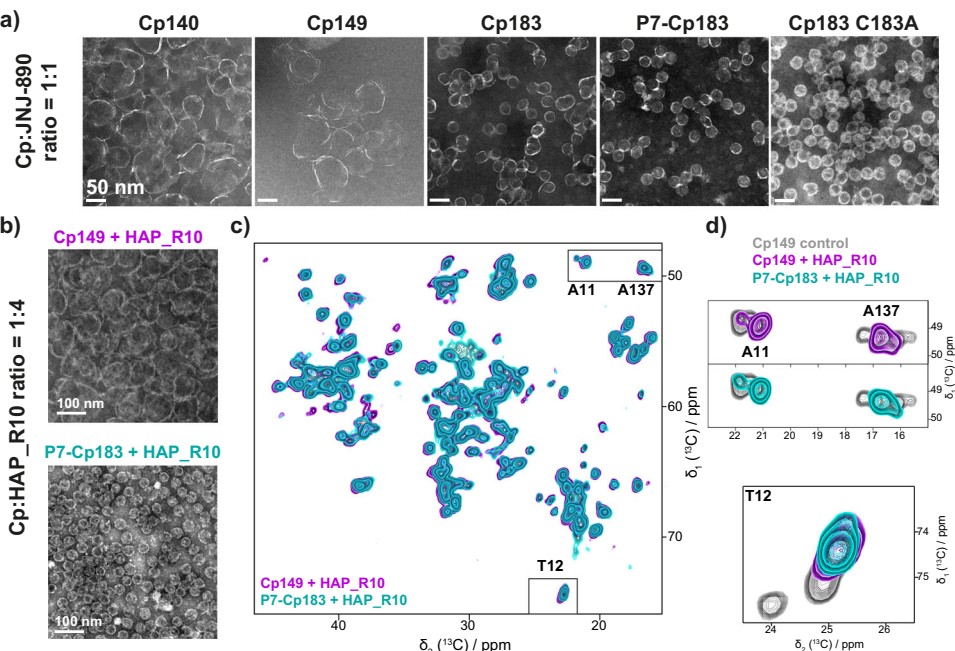

**Fig. 3 | Effect of CAM-A at the mesoscopic and atomic level on truncated and full-length HBV core protein. a** Negative-staining EM micrographs of different capsid constructs after incubation with 1-molar equivalent of the JNJ-890 CAM-A. Capsids made of truncated Cp149 result fully open, while those assembled from full-length Cp183 appear mostly closed. For EM micrographs at four different JNJ-890 ratios see Supplementary Fig. 6. **b** EM micrographs corresponding to the NMR samples of Cp149 and P7-Cp183 bound to HAP_R10. **c** Overlay of $^{13}C$-$^{13}C$ DARR solid-state NMR spectra of Cp149 (purple) and P7-Cp183 (cyan) bound to HAP_R10. **d** Zooms from the DARR spectra showing A11, A137 and T12 resonances, with the reference Cp149 spectrum in gray (from ref. [53]). For EM micrographs, experiments were repeated independently at least 2 times with similar results, except for Cp140 which was done once.

inter-dimer interactions in Cp183 which maintain a roughly spherical structure of the multimer, even if one cannot exclude the possibility that the C-terminal intrinsically disordered regions have an entropic contribution to the capsid assembly without any direct interaction[57]. It is interesting to note also that CAM-A seems to bind preferentially on open assemblies, as can be seen when sub-stoichiometric amounts are added in Supplementary Fig. 6 and Supplementary Fig. 8, where intact capsids are observed next to flat assemblies.

In order to assess whether these differences on the mesoscopic level between open and closed capsids are reflected at the molecular level, we recorded NMR spectra on Cp183 and P7-Cp183 capsids in the presence of JNJ-890 and HAP_R10 CAM-A (Supplementary Fig. 9). We have previously shown that the spectra of the Cp149, Cp183 and P7-Cp183 apo forms virtually coincide[53]. The spectra of Cp149 and P7-Cp183 with HAP_R10 (EM in Fig. 3b, NMR in Fig. 3c, and zooms in Fig. 3d), reveal that this is surprisingly also the case for the HAP_R10-bound capsids. The same observation is made for Cp183 with JNJ-890, as shown in the spectra in Supplementary Fig. 10. Notably, in both Cp183 and P7-Cp183 preparations in presence of CAM-A, a small residual signal could be observed, either stemming from pentamers that might still be present at NMR detectable levels/ratios, or from CAM-free sites, or both.

We realized when analyzing different NMR rotor contents of Cp/CAM-A preparations by EM that 1,4-dithiothreitol (DTT) can enhance capsid opening when used on Cp183 and P7-Cp183. Indeed, while all samples had yielded the same NMR spectra, appearance under EM was quite different, and correlated with the presence of DTT in the buffer. DTT alone had no impact on the NMR spectra of Cp besides reducing the peak intensity for oxidized Cys[53]. Supplementary Fig. 11 shows the impact of DTT on Cp183 capsids in presence of JNJ-890. While Cp183 were largely spherical at a Cp:CAM 1:1 molar ratio, they were heavily deformed or even disassembled when DTT was present, similar to the effect observed on Cp149 + CAM-A in absence of DTT (Fig. 3a).

We subsequently tested whether a straightforward interpretation of DTT opening disulfide bonds, notably a bond involving C183 and C48

as postulated recently[58], or the bonds present in the Cp150 with C-terminal Cys used for cryo-EM[26,34,35,37], might hold, through using different Cys mutants (Supplementary Fig. 11). Interestingly, notably the Cp183 C183A mutant showed no opening in presence of CAM-A, as also observed for wild-type Cp183. This excludes a stabilizing effect of this cysteine as cause for increased capsid integrity, and also confirms that CTD-meditated stabilization is not due to oxidative crosslinking and formation of a covalent intra-capsid network[59]. All other Cp183 cysteine mutants also resisted to disassembly in presence of CAM-A. Upon addition of DTT, Cp183 C183A readily disassembled, while C107A, and to a lesser extent C61A, were least affected by addition of DTT, suggesting that in Cp183 WT the mutated Cys residues are somehow involved in stabilizing the spherical capsid-like structure in a DTT-sensitive fashion, but not by S–S bond formation, which are sterically not possible. However, how this might work on a molecular basis remains unclear, and no simple picture emerged from our experiments.

Another capsid-stabilizing factor in CTD-containing *vs.* CTD-less Cp variants can be packaged RNA[60–62], as its multivalent electrostatic interactions with the positively charged CTDs could also counteract particle disruption. Similarly, interactions between phosphorylated CTDs and positively charged arginine residues in P7-Cp183 can stabilize capsids[63]. Still, how DTT could impact these interactions remains to be determined.

Finally, we assessed the packaged RNA content of *Escherichia coli* Cp183 capsids incubated with both types of CAMs. For this, we compared the $^{13}C$ RNA NMR signals in the 2D DARR spectra recorded on Cp183 capsids with and without CAM (Supplementary Fig. 12). Through its arginine-rich domain, Cp183 can package the equivalent of roughly 4000 nucleotides of *E. coli* RNA[60,62,64]. Both CAM-A compounds reduced the RNA signals by around 25% for JNJ-890, and 50% for HAP_R10 in the aberrant assemblies, which either indicates that RNA was released from the capsid CTDs, or that RNA is more mobile, which would result in reduced dipolar polarization transfers. In contrast, the JNJ-632 CAM-E had no significant impact on the RNA signal

intensity, meaning that the RNA remained protected in the non-disassembled capsids, as expected for CAM-E.

In summary, we conclude that, despite the differences observed by microscopy, Cp149, Cp183 and P7-Cp183 all undergo similar molecular-level changes in response to the JNJ-890 CAM-A. At the mesoscopic level, the opening of capsids in the presence of CAM-A is in a yet unexplained manner assisted by DTT.

## Modulation by CAMs directly during capsid assembly

Directly investigating re-assembly of Cp183 dimers in the presence of CAMs is difficult, as it often leads to aggregation of the protein. As an alternative, we have recently shown that capsid assembly of Cp183 in the presence of mRNA and CAMs can be achieved by using wheat-germ extract cell-free protein synthesis (WG-CFPS)[65]. We here use this approach to investigate whether capsid assembly directly upon Cp183 exit from the ribosome is modulated in the same way as on preformed capsids, and how nucleic acid packaging and phosphorylation are orchestrated in presence of CAM. One should mention that DTT is systematically present in the cell-free reaction[66]. We synthesized isotope-labeled Cp183 capsids in the cell-free system (in the following referred to as CF-Cp183) in the presence of different CAMs. By negative-staining EM of the assemblies directly in the crude CFPS reactions carried out with increasing amounts of CAM-A, we found that a Cp monomer:JNJ-890 ratio of 1:0.4 was sufficient to open most capsids (Supplementary Fig. 13), which was therefore chosen for NMR sample preparation with JNJ-890 and HAP_R10 CAM-A. For the JNJ-632 CAM-E, a 1:1 molar ratio was used (Supplementary Fig. 14). Interestingly, while CF-Cp183 is, as in absence of CAM (Supplementary Fig. 14a), largely found in the soluble fraction in presence of JNJ-632 (Supplementary Fig. 14b), it localizes to a large part to the insoluble pellet fraction with JNJ-890 and HAP_R10 (Supplementary Fig. 14c, d).

The resulting EM micrographs and 1D $^{31}$P and 2D hNH spectra are shown in Fig. 4. As observed with *E. coli*-produced Cp149 and Cp183, JNJ-632 gave rise to mesoscopically normal capsids, whereas both CAM-A compounds induced large aberrant assemblies that appeared more open with JNJ-890 than with HAP_R10 (Fig. 4a). It can be noted that even Cp183 capsids open in the presence of CAM-A when synthesized in the CF system. This is likely due to the presence of DTT in the CF reaction, which we have shown above has an impact on the morphology of Cp183 in presence of CAM-A (see above and Supplementary Fig. 11). In line with the DARR spectra of the *E. coli* samples, the hNH spectra recorded with both CAM-A compounds lacked peak multiplicity, indicating a loss of $T = 4$ icosahedral symmetry (Fig. 4c). Moreover, the spectra from both the bacterial and cell-free samples were virtually superimposable, (Supplementary Fig. 15), indicating the absence of local conformational differences.

We have recently shown that the WGCF system supports phosphorylation of viral proteins[19,67,68], and were therefore able to analyze the phosphorylation state of CF-Cp183 in the presence of CAMs. The $^{31}$P spectrum of CF-Cp183 in absence of CAM indicates that no phosphorylation occurred in these capsids (since no $^{31}$P signal at 6 ppm is observed), while clear signals for Cp-associated RNA were seen around −1.5 ppm (Fig. 4b) (the RNA packaged by the capsid during cell-free synthesis is the mRNA coding for the protein[65]). This result is in line with data for *E. coli*-produced Cp183 capsids[62] which package RNA when unphosphorylated, but not when highly phosphorylated, as in P7-Cp183. The same experiment done with addition of JNJ-632 reveals a similar RNA content as judged by the intensity of the signal, and also absence of phosphorylation. Hence RNA is packaged into capsids in the presence of a CAM-E compound when assembled in the cell-free system, although in mammalian cells pgRNA content was found to be substantially reduced in the presence of other CAM-E compounds, such as AT-61[69]. In contrast, when JNJ-890 was added during cell-free protein synthesis, no RNA signal was detected in the $^{31}$P spectrum, while a signal typical for phosphorylated amino acids was detected

around 6 ppm[53]. The signal for phosphorylation was weaker than the one detected for RNA in the CAM-E samples, despite comparable NMR spectrum acquisition times. Explanations could include a smaller number of $^{31}$P atoms in the phosphorylated proteins than in the equivalent of about 4000 nucleotides of RNA (if the RNA content is similar to the one observed in *E. coli*[60,64]), or a higher mobility of the phosphorylated Cp183 CTD when compared to the RNA phosphate backbone, leading to a less efficient cross-polarization transfer and hence reduced peak intensity in the NMR experiment. Clearly, no rigidly bound RNA can be detected in CAM-A Cp assemblies.

In order to assess the extent of phosphorylation, mass spectrometry of capsids assembled in the presence of JNJ-890 was performed, and revealed eight-fold phosphorylation, and no signal for unphosphorylated protein (Supplementary Fig. 16). The fact that eight phosphorylated sites are observed suggests that there are more kinases present in the WGE than just SRPK1, whose coexpression in *E. coli* results in phosphorylation of seven sites[62]. It is therefore likely that, while the lower number of $^{31}$P atoms per capsid can account for about a factor of two in the loss of signal, at least part of the lower signal intensity indeed comes from the higher dynamics of the CTD relative to the phosphate backbone of the RNA in the RNA-containing capsids. In addition, when HAP_R10 was added, neither phosphorylation nor bound RNA were detected (Fig. 4). This could support the interpretation that either the CTD is too mobile to detect phosphorylation in presence of HAP_R10, or, less likely, that RNA remains loosely bound and is thus rather mobile in the presence of CAM-A, not yielding strong signals in the cross-polarization-type transfers used for the $^{31}$P experiments.

In summary, we observe that capsids assembled in the presence of CAM-E and nucleic acids indeed package the mRNA present in the cell-free synthesis reaction, even while in cells they do not package pgRNA. Capsids assembled in presence of CAM-A on the contrary are fully phosphorylated in this in vitro WG-CFPS set up.

## Discussion

We here present insight into the structural details of aberrant HBV Cp assemblies formed in the presence of CAM-A capsid assembly modulators. We showed that in the resulting assemblies, a single conformation of Cp is present, in contrast to the four distinct, quasi-equivalent Cp monomer conformations in the icosahedral $T = 4$ capsid. We further demonstrate that the NMR fingerprint directly identifies residues involved in CAM binding, allowing to use NMR chemical-shift perturbations as discriminators between CAM-E and CAM-A. In addition, our data reveal that, despite the different appearance on the mesoscopic level as seen in electron micrographs, the assembly domains of full-length Cp183 and P7-Cp183 adopt the same structure as Cp149 on the molecular level when bound to CAM-A. We identified how different CAMs modulate Cp183 phosphorylation and RNA packaging upon Cp assembly directly on exit from the ribosome in a cell-free protein synthesis system.

How do capsids with four different molecules in the asymmetric unit transit to objects with only a single conformation when bound to CAM-A? First, one should mention that, as in Cp capsids, crystalline Ubiquitin[70–73] and SH3[74] show peak multiplicity related to the number of molecules in the asymmetric unit. This is thus a feature also observed for other proteins. Still, $T = 3$ icosahedral Nackednavirus[75] capsids do not show the expected peak multiplicity (manuscript under review), but reveal substantially broader NMR signals than the typically very narrow signals of Ubiquitin, GB1 and the HBV capsid, pointing to small, unresolved peak splitting instead of a clear multiplicity. That Cp assemblies in the presence of CAM-A yield very narrow NMR signals, without any sign of multiplicity indicates that, within the limit of the (narrow) line widths, a single molecule is present in the asymmetric unit of the aberrant assemblies formed in the presence of CAM-A. Still, smallest heterogeneities might only be observable if NMR lines would

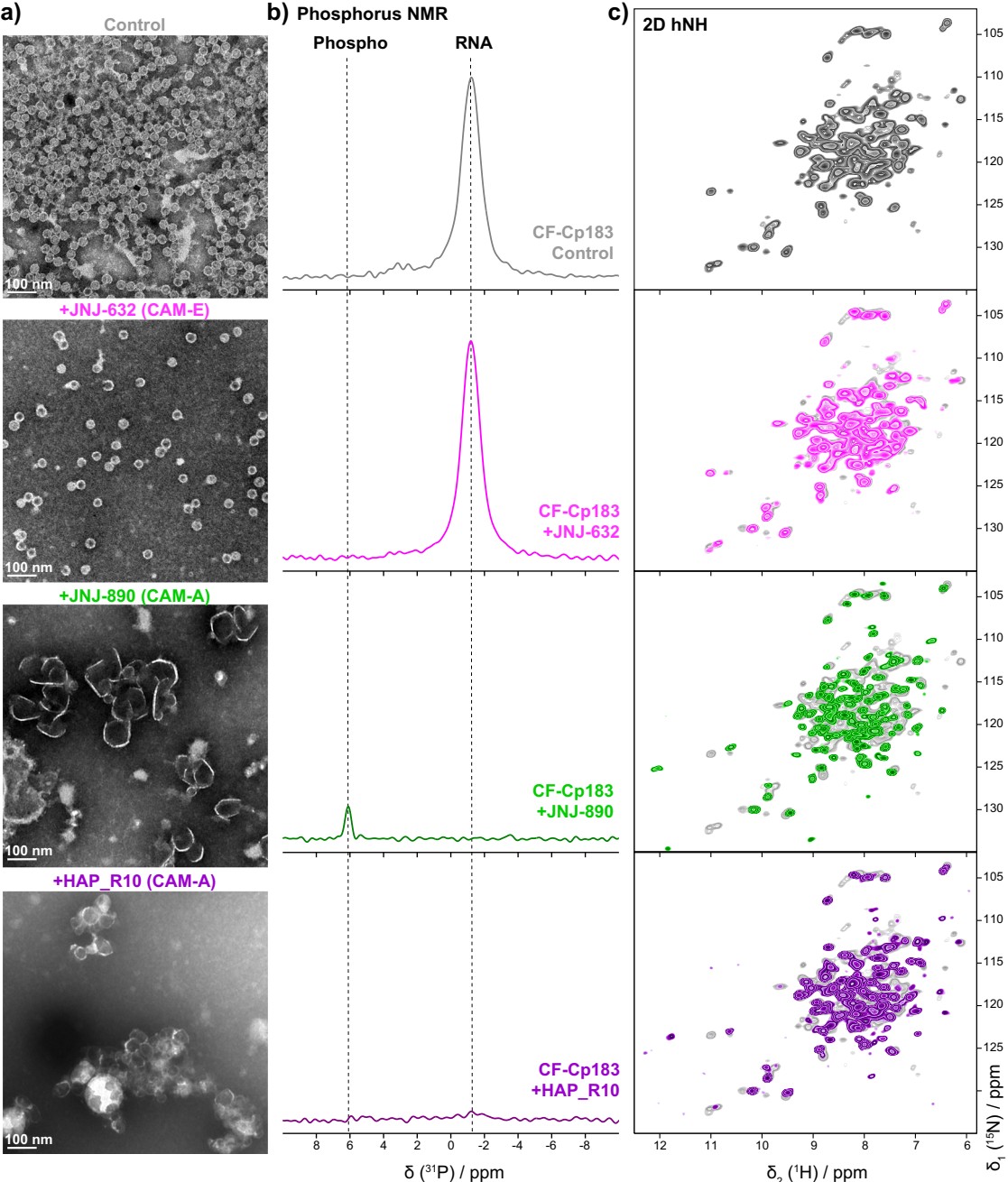

**Fig. 4 | Effect of CAM-A and CAM-E on Cp183 CTD phosphorylation and RNA packaging. a** Negative-staining EM micrographs of NMR samples of Cp183 from WG-CFPS as control. EM micrographs were taken on resuspended sediments after 1.3 mm rotor filling (see corresponding SDS-PAGE in Supplementary Fig. 13). $n = 1$ independent experiments have been recorded. **b** 1D $^1$H-$^{31}$P CP spectra of cell-free synthesized Cp183 (CF-Cp183) without CAM (gray), assembled in presence of 1

molar equivalent of JNJ-632 (pink), and 0.1 molar equivalent of JNJ-890 (green) and HAP_R10 (purple) recorded at 55–60 kHz MAS on a 500 MHz spectrometer. **c** 2D hNH spectra of the corresponding samples recorded at 60 kHz MAS on a 800 MHz spectrometer. Spectra of the control sample are shown in light gray for comparison. Corresponding CSPs are shown in Supplementary Fig. 15.

be even narrower. If the $T = 4$ capsids simply opened to form planar lattices, they would transit into a planar trihexagonal tiling showing three different molecules in the asymmetric unit (B, C and D monomers). Since the spectra show only a single signal, a single molecule must be present in the asymmetric unit, as in a hexagonal lattice shown in Fig. 5b. That the aberrant Cp149 assemblies are not fully planar might be due to the residual presence of pentameric vertices which remain below the detection limit at the current signal-to-noise ratio, i.e. <5%. This would however suggest that less than one pentameric vertex ($12 \times 5\% = 0.6$) appeared in every particle. This seems however unlikely based on the observed micrographs. It appears more likely to

us that inter-dimer interactions in the presence of CAM-A in WT capsids are not compatible with a fully planar arrangement. The single signals also indicate full occupancy with CAM-A, since no (or only very weak in some preparations) NMR signals corresponding to the unbound state are observed. This is in contrast to cryo-EM studies of Cp150 capsids which revealed binding of one CAM-A per dimer[26,35,37]. The partial occupancy there might be due to the forced maintenance of a closed capsid structure by disulfide crosslinking of the C-terminally added cysteine[34], while the planar symmetry in the Cp149 Y132A crystals sterically allowed saturation of all sites[39]. One can mention that most residues are more flexible in the presence of CAM-

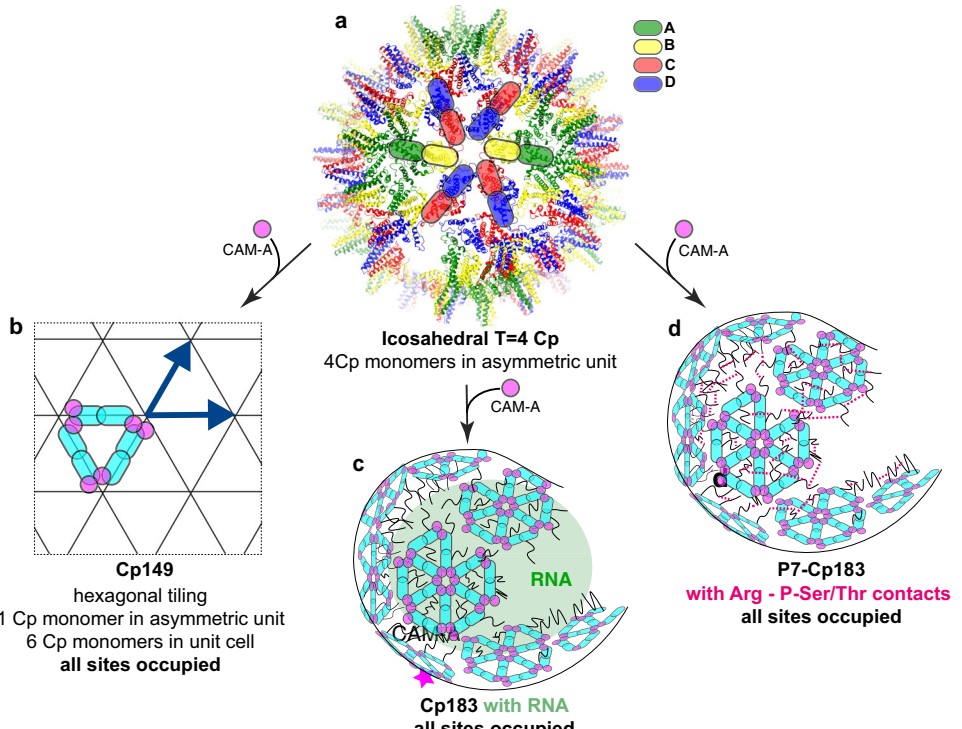

**Fig. 5 | A molecular model for capsid assembly modulation. a** Capsid without CAM. **b**, **c** Models of capsid and dimer rearrangements in the presence of CAM-A. **b** The hexagonal tiling resulting from capsid opening and loss of the pentameric sites is shown with all molecules colored in cyan; CAM-A are shown in pink. **c** Effect of CAM-A on Cp183 capsids. All CAM binding sites are occupied, and pentameric axes are abolished. RNA (green) holds the Cp dimers in an approximately round shape. **d** Effect of CAM-A on P-Cp183 capsids, forming similar superstructures to the capsids in c, but with Arg-P-Ser/Thr interactions (deep pink broken lines) stabilizing the capsids.

A. This globally increased flexibility points to a less constrained assembly, where notably the spike can make larger excursions.

A typical NMR fingerprint is observed for CAM-interacting Cp. The involved amino-acid residues in part coincide with those identified previously[6], but the NMR chemical-shift analysis interestingly adds the region spanning residues 14–17, and identifies in greater detail which residues are involved in the region spanning residues 124–134.

Most intriguingly, we observe different morphology for Cp149 and Cp183 in the presence of CAM-A on the mesoscopic scale: while the truncated Cp149 capsids open up at much lower CAM-A concentrations, Cp183 capsids retained a virtually globular shape even at high CAM concentrations. This observation becomes central in the context of a recent study in full-length Cp183-expressing Huh7 cells, which investigated a CAM-A, BAY 41–4109[23]. In this cellular context, CAM-A did not induce the irregular assemblies generally seen by EM for Cp149 in vitro, but instead dense arrays of what resembled intact capsids. The authors concluded that this might reflect a different mechanism of BAY 41–4109-mediated Cp misassembly under in vivo *vs.* in vitro conditions[23]. Our data at the mesoscopic level corroborates their findings, but the NMR data demonstrate that the effect of CAM-A on Cp149 and Cp183 is, despite the apparent difference, the same at the molecular level. According to our findings, negative-staining electron micrographs used as readout are not resolutive enough to reflect that, while the capsids seem intact, they actually lost the distinctive icosahedral symmetry of intact capsids (Fig. 5c). The seemingly intact shape/morphology of CAM-A treated Cp183 but not Cp149 capsids at a mesoscopic scale is most likely maintained by intermolecular interactions involving the CTD of Cp183, as sketched in Fig. 5c, d for Cp183 and P-Cp183 respectively. This is plausibly explained by RNA-CTD interactions in non-phosphorylated capsids, and has also been observed in a previous EM study, where empty Cp183 capsids were destabilized by the synergic action of Importin-β

and HAP12, but RNA-filled capsids resisted[47]. However, CTD-RNA interactions cannot account for the high stability of P-Cp183 capsids which are devoid of RNA. As alternative intermolecular interactions involving the CTDs the positively charged arginine side chains can form salt-bridges with the phosphorylated sidechains from other P-Cp183 dimers, building a densely woven interaction network which maintains the apparently globular particle shape in a similar way as when interacting with packaged RNA. This view is meanwhile experimentally supported by recent NMR measurements which show that the CTDs of P7-CP183 show a similar dynamic behavior to those of pgRNA-filled Cp183, and must thus be involved in interactions, likely of intermolecular nature, between arginine side chains and phosphorylated Ser/Thr residues[63].

Intriguingly, the apparent stability of Cp183 particles against CAM-A was reduced upon addition of DTT. Current evidence points to a role for Cys107 and, to a lesser extent, Cys61 although the underlying mechanism is as yet unclear. Still, these observations might be of importance for investigations on co-factors of CAM-A-induced capsid destabilization, as recently reported for importin-β[47]; the presence of DTT in such experiments might induce unexpected effects. While the oxidation state, notably of C61, has shown to impact capsid stability[76], it remains an open question today how it can induce at the molecular level the more readily observed capsid opening in presence of CAM-A both under reducing (DTT) or oxidizing (longer time delays) conditions.

Capsid assembly modulation directly upon exit of Cp183 from the ribosome in a cell-free protein synthesis system revealed how CAMs impact Cp183 capsid assembly in situ and in the presence of RNA. Interestingly, RNA packaging was distinctly affected compared to reported CAM activities in live mammalian cells. There, CAM-E compounds interfere with pgRNA encapsidation[22,69,77,78], whereas the [31]P NMR spectra in our work clearly revealed packaged mRNA in capsids

from cell-free synthesized Cp183 in the presence of CAM-E. It is indeed difficult to apprehend how CAMs could actually inhibit packaging of RNA on capsid formation, since charge equilibration seems an essential driving force in capsid assembly, and the high concentration of positively charged arginine residues in the CTD makes this process in the absence of RNA difficult under physiological conditions. The empty capsids described to form in cells in the presence of CAM-E must thus either be phosphorylated to a large extent, or they contain nonspecifically packaged cellular RNA, which would typically not have been detected in the corresponding studies, where pgRNA sequence-specific hybridization probes were used[69,77].

In contrast, in CAM-A treated cell-free synthesized Cp183 samples did not show any signal for Cp183-associated RNA, conceivably because the resulting flat assemblies (due to the presence of DTT) cannot act as protective RNA containers, in contrast to the preformed *E. coli* capsids treated with CAM-A, which maintain more than 50 % of the packaged RNA (Supplementary Fig. 12). Interestingly, however, cell-free synthesized Cp183 assembled in presence of JNJ-890 gave a weak, but clear signal of protein phosphorylation, supported by the presence of adequate kinases in the wheat-germ extract used for cell-free protein synthesis[68], whereas no phosphorylation was seen in HAP_R10-treated and neither in untreated Cp183 samples. How the molecular recognition of the viral genome, which has been shown to be dependent on the quasi-equivalence in the asymmetric unit[79], would be impacted by CAMs remains to be established.

To conclude, we here used solid-state NMR to show that the molecular level organization of the aberrant HBV Cp assemblies induced by modulation with CAM-A compounds changes from $T = 4$ icosahedral to hexagonal. The fact that the mesoscopically irregular assemblies actually show high order on the molecular level, as reflected in the narrow lines in the NMR spectra, was surprising, and shall further motivate NMR investigations of systems which have a mesoscopically heterogenous appearance. We showed that NMR fingerprints can clearly distinguish between CAM-A and CAM-E type of action, even for full-length Cp183 capsids where EM images can be misleading. Importantly, our data reconcile in vitro and in vivo views on CAM-A mode of action. Our analyses also reveal an influence of DTT on aberrant capsid formation, but the underlying principles remain not fully understood. Using a combination of advanced synthetic biology and NMR techniques, we also elucidated how CAMs affect phosphorylation and RNA packaging in vitro during Cp183 capsid assembly, challenging previously established interpretations of empty capsid formation by CAM-E.

In summary we showed that NMR fills important gaps in the understanding of the molecular effects of notably CAM-A, and by this contribute to the further development of capsid assembly modulation.

## Methods

### Capsid assembly modulators (CAMs)
GS-832471 and GS-942049 are from Gilead. JNJ-632, JNJ-827 and JNJ-890 are from Johnson & Johnson. GS-837886 (called HAP_R10 here) was a kind gift from Roche. All compounds were dissolved in 100% DMSO to a final concentration of 10 mM and stored at −20 °C. Formula and molecular weights of all compounds are shown in Supplementary Table 1.

### Cp from bacterial expression and incubation with CAMs
Plasmids were transformed into *E. coli* BL21* CodonPlus (DE3), grown at 37 °C in M9 in media as described below. When the optical density at 600 nm (OD600) reached between 0.7 and 2.0, expression of Cp140/149 was induced with 1 mM isopropyl ß-D-1thiogalactopyranoside (IPTG) overnight at 25 °C. For full-length proteins (Cp183 WT and mutants as well as P7- Cp183), expression was induced at an OD600 of 2.0 overnight at 20 °C. Cells were collected after centrifugation (20 min at 6000 × $g$) and resuspended in TN300 buffer (50 mM Tris, 300 mM NaCl, 2.5 mM ethylenediaminetetraacetic acid (EDTA), 5 mM DTT, pH 7.5). Cell suspensions were incubated under addition of 1 mg/mL chicken lysozyme,

1× protease inhibitor solution, and 0.5% Triton X100 (TX-100) (45 min). Six microliters benzonase (for 1 liter of culture) was added for 30 min at RT. After sonication (10 cycles of 10 s and 50 s cooling), cell lysates were centrifuged (8000 × $g$) for 1 h. The supernatant was loaded into a 10 to 60% sucrose gradient (in 50 mM Tris pH 7.5, 300 mM NaCl, and 5 mM DTT) and centrifuged using a SW-32Ti (Beckman Coulter) swinging bucket rotor at 140,000 × $g$ (3 h at 4 °C). The presence of capsids was identified by SDS-PAGE. Sucrose fractions containing capsids were precipitated by saturated (NH4)2SO4. After centrifugation at 20,000 × $g$ (1 h), pellets were resuspended in 10 mL buffer (50 mM Tris pH 7.5, 5% sucrose, 5 mM DTT, and 1 mM EDTA). The protein solution was centrifuged again for 15 min to remove insoluble pellet. For EM studies, all Cp samples (Cp140, Cp149, Cp183, P7-Cp183 and Cp183 C183A mutant) were expressed in *E. coli* using LB medium and purified as described above. Samples were dialyzed in 50 mM HEPES buffer at pH 7.5 and kept at 4 °C. For solid-state NMR studies, Cp149, Cp183 and P7-Cp183 were expressed in *E. coli* using M9 medium supplied with 15NH4Cl and 13C-glucose, and purified as described above. Cp capsid samples were incubated at a concentration of ~1 mg/ml (corresponding to ~60 μM monomer concentration for truncated Cp140 and Cp149 and ~46 μM for full-length Cp183 and P7-Cp183) with relevant monomer:CAM ratio for 2 h at 37 °C in 50 mM HEPES buffer at pH 7.5, in presence of 5 mM DTT when specified.

### Preparation of solid-state NMR samples produced in *E. coli*
13C-15N-labeled Cp183 and P7-Cp183 capsids produced in *E. coli* were dialyzed in 50 mM TRIS at pH 7.5 and loaded onto a HiPrep™ 16/60 Sephacryl® S-200 HR column (120 ml) to remove Triton-X100 which was shown to bind in the hydrophobic pocket of Cp and induces large CSPs in the NMR spectra1, except for the samples of Cp183 with JNJ-632 and JNJ-890. For each sample, between 15 and 20 mg of autoassembled capsids at a concentration ~1–1.5 mg/ml were incubated at a monomer:CAM ratio of 1:4 for 2 h at 37 °C in 50 mM TRIS buffer at pH 7.5 in presence of 5 mM DTT. In the case of Cp149, NMR samples were prepared starting from Cp dimer (see Supplementary Table 2). For this, 13C15N-labeled and 2 H13C15N-labeled Cp149 capsids were disassembled as described in[53]. Freshly separated dimers after gel filtration were dialyzed in 50 mM HEPES buffer at pH 7.5, 5 mM DTT. Concentration was determined using a nanodrop measurement, adjusted to ~1 mg/ml (~30 μM dimer concentration), and dimers were incubated with a monomer:CAM ratio of 1:4 for 24 h at room temperature (RT) in presence of 150 mM NaCl. Reassembled capsids samples were diluted by 2 in 50 mM HEPES buffer at pH 7.5 to reduce the final NaCl concentration to 75 mM, since high salt concentrations may reduce the NMR sensitivity. For all NMR samples, final DMSO content always stayed below 3%, and EM grids were prepared at the end of the incubation step. Samples were concentrated to 1 ml final volume and sedimented into 3.2 mm rotors by an overnight ultra-centrifugation at 200,000 × $g$, 4 °C. Excess of sediment was removed and 1.5 μl of saturated DSS (0.25 M) were added before closing the rotor for NMR chemical shifts referencing. In addition, 2 H13C15N-labeled Cp149 capsids in presence of JNJ-890 were filled in 1.3 mm and 0.7 mm rotors for 1 H-detection NMR experiments and comparison with cell-free samples. To validate that the Cp149/CAM complexes did not disassemble upon NMR sample preparation, and notably during the ultracentrifugation step, EM micrographs were also taken on excess of sediment resuspended in 50 mM HEPES buffer at pH 7.5 on representative samples (Cp149 with JNJ-632, JNJ-827 and JNJ-890) (Supplementary Fig. 4b), and showed similar objects as prior the ultracentrifugation as shown in Fig. 2a. A summary of all the NMR samples prepared is shown in Supplementary Table 2.

### Cp from wheat-germ cell-free protein synthesis
Cp proteins have been produced in the wheat-germ cell-free system using the dialysis method, also called CECF for continuous exchange

cell-free system[66,80]. Small-scale experiments have been performed in CECF-mini-reactors manufactured at ETH Zurich after models from reference[81], while medium-scale and large-scale experiments have been performed in 500-μl and 3-ml dialysis cassettes, respectively[50,82]. In all cases, transcription and translation have been performed separately. In mini-reactors, the translation mixture contained ½ volume of home-made wheat-germ extract (WGE), ½ volume of mRNA, 40 ng/μl of creatine kinase and 6 mM of amino acid mix for a total volume of 70 μl, while in dialysis cassettes, it was composed of ½ volume of feeding buffer, 1/3 volume of mRNA, 40 ng/μl of creatine kinase, 0.3 mM of amino acid mix, and 1/6 volume of homemade WGE. The feeding buffer (1.5 ml for a minireactor, and 20 ml or 120 ml for a 500-μl or a 3-ml dialysis cassette, respectively) contained 30 mM HEPES-KOH pH 7.6, 100 mM potassium acetate, 2.7 mM magnesium acetate, 16 mM crea-tine phosphate, 0.4 mM spermidine, 1.2 mM ATP, 0.25 mM GTP and 4 mM DTT (SUBAMIX NA, CellFree Sciences), supplemented with 6 mM of amino acid mix. A mix containing all twenty isotopically labeled amino acids (Cambridge Isotope Laboratory) was used for the pro-duction of 2 H13C15N-Cp183 in a 3-ml-translation reaction experiment for NMR studies. A dialysis membrane with a molecular weight cut-off of 10 kDa has been used, and protein synthesis has been performed for 16 h at 22 °C under agitation. CAMs have been added both to the translation mix and to the feeding buffer at different molar ratios, considering the Cp concentration to be 0.25 mg/ml WGE. The total cell-free reaction mixture for CF-Cp183 and CF-Cp183+JNJ-632 was treated with 25,000 U/ml of benzonase for 30 min at room temperature before centrifugation at 20,000 × g, 4 °C for 30 min. The supernatant (SN) was loaded onto a discontinuous sucrose gradient with layers of 10 to 60 % as described in 7, and 50–60% fractions were used for 1.3 mm NMR rotor filling by overnight ultracentrifugation (200,000 g, 4 °C). No benzonase treatment was done when CAMs-A were added to the reaction, and the pellet fractions were used for 1.3 mm NMR rotor filling. For Western Blotting analysis, a polyclonal rabbit antiserum against the assembly domain of the HBV core protein (a-c149) was used for detection of Cp183 (Eurogentec, custom antibody).

### Negative-staining electron microscopy

For EM grids preparation, 5 μl of sample were adsorbed to the surface of carbon-coated grid for 2 min, sample excess was removed with paper, and grids were stained with 2% uranyl 5 acetate for 2 min. Samples were imaged with a JEM-1400 transmission electron micro-scope operating at 100 kV.

### NMR experiments for chemical-shift analysis and dynamics measurements

NMR experiments were conducted using a 1.3 mm and a 3.2 mm triple-resonance ($^1$H, $^{13}$C, $^{15}$N) probe heads at static magnetic field of 18.8 T corresponding to 800 MHz proton resonance frequency (Bruker Avance III). Assignments of unbound Cp were taken from ref. [49] (BMRB number 27317) and [53] (BMRB number 28122). Cp-CAM-bound resonances were assigned using a combination of 2D and 3D correlation experiments including: 2D DARR, 2D NCA, 3D NCACX, and when necessary a 3D CANCO[83,84]. Carbon-detected experiments were recorded at a MAS frequency of 17.5 kHz on all samples and are detailed in Supplementary Table 3. Proton-detected experiments were recorded at a MAS frequency between 55 and 60 kHz and are detailed in Supplementary Table 4. All spectra were referenced using DSS and recorded at a sample temperature of 4 °C in the 3.2 mm probe and between 20 and 25 °C in the 1.3 mm probe, as determined by the resonance frequency of the supernatant water[85]. For dynamics measurements, site-specific solid-state NMR relaxation-rate constants $R_{1\rho}(^{15}N)$ were measured on $^2$H$^{13}$C$^{15}$N Cp149 alone and in interaction with JNJ-890, using the 3D hCANH sequence described in reference[19] in an 0.7 mm rotor at 80 kHz MAS, 20 T (850 MHz) external magnetic field strength and with a 13 kHz spin-lock field. Eight 3D hCANH

experiments were recorded with the spin-lock delay time varied between 1 μs and 251 ms. For further experimental information see Supplementary Table 5. The obtained 3D hCANH spectra were assigned using published assignments[49] and resonances intensities are extracted using CcpNmr[86,87]. Mean site-specific relaxation-rate constants were extracted from the 3D hCANH experiments within MATLAB using the spectral fitting package INFOS[88]. Relaxation curves were fitted with a mono-exponential fit with two degrees of freedom $(A \cdot \exp(-Rt))$ and are shown for all resonances in Supplementary Fig. 3c in the absence of JNJ-890 and in Supplementary Fig. 3d in the presence of JNJ-890. Experimental errors σ for the relaxation-rate constants have been determined using bootstrapping with 200 iterations for relaxation time data. All error bars are given twice the standard deviation (σ). $^{31}$P,$^1$H cross-polarization experiments were recorded at 11.7 T (500 MHz proton frequency) in a triple-resonance 1.3 mm probe. The MAS frequency was set to 60 kHz and the sample temperature was set between 20 and 25 °C. The CP experiments were optimized on solid $NH_4H_2PO_4$ and typical radio-frequency field strengths used in the CP were 100 kHz for $^1$H and 40 kHz for $^{31}$P. The CP contact time was set to 1 ms and the repetition time to 1.5 s. The acquisition time was set to 10.2 ms. WALTZ-64 $^1$H decoupling (5 kHz rf-field strength) was applied during acquisition. The number of scans was 52240 (CF-Cp183 and CF-Cp183 + JNJ-632), 41984 (CF-Cp183 + JNJ-890) and 58408 (CF-Cp183 + HAP_R10). Note that the spectrum of CF-Cp183 was recorded at 55 kHz MAS and thus slightly different CP conditions. All spectra were referenced to $H_3PO_4$. Processing of all spectra was done using TopSpin 3.2 (Bruker Biospin) with zero filling and a squared cosine apodization function (SSB 2.5–3 depending on spectrum and dimension). Spectra analysis and assignment was done with the CcpNmr Analysis package[86,87]. Chemical-shift perturbations induced by CAMs were calculated for all $^{13}$C and $^{15}$N assigned nuclei and plotted using a homemade python script (Eq. 1). For $^{15}$N nuclei, the CSPs were calculated by using a scaling factor of 0.4 for the car-bon to take into account the smaller chemical-shift range (when considering Cα) (Eq. 2):

$$\triangle\delta_N = 0.4*|\delta_N[\text{bound}] - \delta_N[\text{unbound}]| \qquad (1)$$

$$\triangle\delta_{HN}\sqrt{(\triangle\delta_H)^2 + (0.1*\triangle\delta_N)^2} \qquad (2)$$

### Reporting summary
Further information on research design is available in the Nature Portfolio Reporting Summary linked to this article.

## Data availability
The data that support this study are available from the corresponding authors upon reasonable request. The CPSs and relaxation source data underlying Fig. 1f and Supplementary Figs. 1b, 3a, 5a, and 15 are pro-vided as Source Data file. Fits for individual relaxation decays are given in the SI. NMR spectra are available upon request. Reference spectra in Figs. 1–3, and Supplementary Figs. 4, 9, 10 and 12 were published in Lecoq et al. PNAS 2021, as referenced. Previously published PDB codes referred to in this manuscript are 1QGT; 5D7Y; 5WRE. Source data are provided with this paper.

## Code availability
The code for the relaxation data analysis is available in the SI.

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

## Acknowledgements

This work was supported by the CNRS (CNRS-Momentum 2018, L.L.), the French Agence Nationale de Recherches sur le Sida et les hépatites virales (ANRS, ECTZ71388, & ECTZ100488, both A.B.; L.B. ANRS ECTZ158948), and a PhD grant from the Chinese Scientific Council to S.W. Financial support from the IR-RMN-THC Fr3050 CNRS for conducting the research is gratefully acknowledged (A.B.). B.H.M. was supported by an ERC Advanced Grant (grant number 741863, Faster, B.H.M), and by the Swiss National Science Foundation (200020_188711, B.H.M). We thank the Centre d'Imagerie Quantitative Lyon-Est (CIQLE) for support at the EM platform (A.B).

## Author contributions

A.B., L.L., M.N. and B.H.M. conceived the study and designed the experiments. L.L. recorded $^{13}$C-detected and $^{1}$H-detected NMR spectra and analyzed NMR data, together with L.B. R.H., S.W. and M.D. prepared capsid samples from *E. coli* expression. L.B. and S.W., together with M.L.F., prepared cell-free synthesized capsid samples. R.H., L.B. and M.B. took EM micrographs. T.W. recorded $^{31}$P spectra. A.M. and M.C. recorded the relaxation data and did the dynamics analysis. A.B. and M.N., together with L.L., B.H.M., and D.D. designed the study. D.B. and J.M.B. selected and provided the CAM compounds. A.B., M.N., B.H.M., L.L. and M.L.F. analyzed data and wrote the manuscript. All authors read and approved the final version of the manuscript.

## Competing interests

The authors declare no competing interests.
