## [Peer Review File · Nature Communications]

Molecular elucidation of drug-induced abnormal assemblies of the Hepatitis B Virus capsid protein by solid-state NMRReviewers' Comments:

Reviewer #1:

Remarks to the Author:

NCOMMS-22-35774

The manuscript of Lecoq et al describes in details the effect of capsid assembly modulators (CAMs) on different forms of the HBV capsid protein. This is an excellent working showing the power of solid-state NMR (SSNMR) to reveal structural details on native systems without reverting to mutant or truncated protein systems. While the effect of CAMs has been studied before by X-ray and cryoEM, those studies required mutants, truncated proteins and conditions that alter the performance of CAMs. In this study, the authors show how full-length HBV capsids (cp183) resist capsid disassembly in the presence of CAM-A unlike cp149 or the presence of DTT present in cryoEM sample preparations. Moreover, large aberrant changes observe in EM images, reproduced by adding DTT, are shown to be associated only with a transformation of symmetry but not of capsid structure. Similarly deformation of the truncated forms do not alter the atom structure of the Cps. Another key finding is that unlike prior suggestions CAM-E do not promote the generation of empty capsids but either undergoes phosphorylation or packages non pgRNA molecules that are required for capsid stabilization. These, and others, are key finding un-route to viral drug design, and also point to some key aspects usually overlooked in such studies, and that refer to the validity of studying such systems not using the original native protein forms.

Methodology is sound, results are well documented an explained, and conclusions are well supported by the data. All experimental work is well documented and can probably be reproduced given the correct expertise in sample preparation and NMR experiments and equipment. One "unresolved" issue is the fate of RNA amount upon addition of CAM-E. A solution for that is suggested in the comments below.

There are several comments that authors have to address prior to publication.

(1) Page 5 – please explain what is the spike part of cp, and its importance. It is later suggested that enhancement dynamics in this region is a dominant drug effect.

(2) A. CSP studies across the manuscript: How are ^{13}C CSPs calculated given there are several carbons per amino acid. Only Ca? RMSD of all assigned atoms? Some normalization factor? What is actually used for the plots in Figure 2c for example?

B. In the SI there are tables with CSP values per atom. There is no meaning for using four digits there. Please reduce the number to the correct accuracy of SSNMR measurements (one digit probably correct)

C. The ^{15}N CSPs are scaled by the gamma ratios. I don't understand the rational. The chemical shift range of nitrogen is ~ 40 ppm (100-140). That of carbon for example is larger, and that of ^1H smaller. Previous studies in the field used values of 0.25, or even different values for different amino acids depending on the BMRB distributions of chemical shifts. When using H-N CSPs, the contribution of ^{15}N is scaled by 0.01, making them unimportant (my guess is the CSP for H alone, and CSP for both will be equivalent). Please elaborate / correct if needed.

D. How do you calculated CSPs when you have split peaks? What is used?

(3) In two cases the authors deduce that reduced line intensities are either from reduction in the amount of RNA or dynamics – from ^{13}C DARR experiments and from ^{31}P CP experiments.

A. Can the authors measure ^{13}C INEPT spectra to assess if the RNA indeed becomes dynamic? At least some of the ribose peaks are unique and will not overlap in 1D INEPT with protein peaks.

B. Regarding the ^{31}P spectra – the authors need a really large amount of scans to get a 1D signal (<40000) which I find strange. Did they try single-pulse excitation to obtain ^{31}P signals? This way the spectrum will be quantitative and it will be possible to determine if the source of signal reduction is dynamics or reduced amount of RNA. With such simple experiments (^{13}C INEPT, ^{31}P 1D) a key question in the article will be answered explaining more about the mode of action of CAM-E.

(4) Another solution to obtain better RNA signals can probably be reduction in temperatures, again confirming the question of dynamics vs. amount of RNA.

(5) Page 20 – why would some CAM inhibit phosphorylation (e.g. HAP_R10) while others (JNJ-890) not in the WG media? Do they inhibit the kinases? This can be checked by production a different

protein in their presence.

Comments on SI:

- (6) Preparation of ssnmr samples – cp183 with JNJ-632 and 890 where not cleared of Triton-X100. Is that because it is not there? It doesn't affect the spectra? Please explain. Can this have any impact on the results?
- (7) Figure S1 is the only one showing some peak splitting in the NH spectra for A137 and S35. The different peaks have completely different intensities. Why?
- (8) Figure S1 – can the H-N spectrum really resolve > 100 peaks as analyzed in (b)? It is difficult to see. Can you show one figure with 'x's marked on all the identified peaks?
- (9) Figure S3: where are the results plotted on pdb 1qgt? There is one plot with the HAP18 but not the plot on free Cp149 ("left"). Also please fix "kJz".
- (10) Figure S5a – same question as above for 13C CSPs. Please explain what we are seeing (I guess this is related to the black bars inside each colored bar, but this is not explained in the caption).
- (11) Figure S15: second "green" is "pink"? What is considered large for HN CSP and what is small? (in numbers)
- (12) Table S2: missing "field" value
- (13) Tables S2: You seem to have performed DQ-DCP for N-C transfer at low spinning rate. This is very uncommon. Can you elaborate a bit? Give a reference? Show an example why this is better than SQ-DCP?
- (14) Table S3: Please make sure fonts are consistent.
- (15) Table S5: as written above – please correct significant figures.

Reviewer #2:

Remarks to the Author:

The manuscript "Molecular elucidation of drug-induced abnormal assemblies of the Hepatitis B Virus capsid protein by solid-state NMR" by Lauriane Lecoq addresses the off-path assembly of HBV nucleocapsid elicited by the Capsid Assembly Modulators (CAM). The mechanism by which CAM work is unknown and the authors used solid state NMR to get insights into this structural off-pathway capsid assembly. The manuscript is scientifically sound and merits publication.

Major points

1. The effect of CAM (CAM-A and CAM-E) seems to show an interrelationship between anomalous assembly and the incorporation of nucleic acid. Both CAMs interact at the same site and led to the breakage of the quasi-equivalence in the asymmetric unit. In my point of view, it is this asymmetry that controls the correct assembly and at the same time the nucleic acid interaction. This manuscript showed unambiguously this effect, but it is lacking a discussion relating the relationship between symmetry in the nucleocapsid, quasi-equivalence in the asymmetric unit and molecular recognition of the virus genome.
2. The authors state that "additional inter-dimer interactions in Cp183 which maintain a roughly spherical structure of the multimer". Although this is valid and even the most probable explanation, but one cannot exclude the possibility that the C-terminal IDR have an entropic contribution to the capsid assembly without any direct interaction. Especially when symmetry/asymmetry is involved in the assembly process. There are many studies correlating entropy and symmetry.
3. In the discussion section, the authors state that "single molecule is present in the asymmetric unit of the aberrant assemblies formed in the presence of CAM-A". Does the authors are considering the possibility of an aberrant particle with T=1. My feeling is that this would be unlikely. It would be more likely to have looser contacts induced by CAM-A, breaking the asymmetry in the asymmetric unit, what would make the icosahedral symmetry unlikely.
4. The effect of DTT on the Cp183 remained unclear. The authors made it clear by the mutant C183A that C183 is not involved in disulfide bond. However, it was not clear if DTT directly interacted with the core protein. Was there any CSP upon addition of DTT?

5. In the experiments of cell free capsid assembly, it would be interesting to see the presence of the dimers or tetramers of Cp183 in solution. Studies have shown that the assembly process may obey the law of mass action of thermodynamics, contributing to the total entropic process (2nd law). Do the authors have any measurements of the relative concentration of dimers, tetramers and capsids in the experiments described in Figure 4 and S14?

6. There are no R1rho information (gaps) for the residues near the described CAM binding site. They are lacking assignment or in conformational exchange?

Minor points

1. Figure S3 – While it is clear in the text, the legend does not state the difference between free and bound.

2. It would be helpful to the reader to have a Supplementary table with CAM-E and CAM-A compounds

Reviewer #3:

Remarks to the Author:

The manuscript entitled “Molecular elucidation of drug-induced abnormal assemblies of the Hepatitis B Virus capsid protein by solid-state NMR” by Lecoq et al reports using solid-state NMR to study various E Coli expressed, pre-assembled HBV capsids interacted with different types of capsid assembly modulators (CAM). They found that CAM-A (formation of aberrant assemblies) induced aberrant structures, despite in an opened conformation, present one single conformation of the capsid protein. This is different to the apo HBV capsids that contain four different conformations of the capsid protein subunit in the icosahedral contact. Furthermore, the authors found the assembly domain of the full-length Cp183 capsid (either unphosphorylated or fully phosphorylated) shows the same conformation at the molecular level to the C-terminal truncated Cp149 capsid when treated with CAM-A despite they looked quite different under the negative stain TEM. Finally, the author examined the effect of CAM to the capsid protein subunit during the assembly process using cell-free translation system.

Overall, the manuscript is very interesting and it represents an important addition to the field. It is of general interest for the readership of Nature Communication. However, I also found some results would need to be further clarified before the manuscript can be accepted for publication.

1. Based on the negative stain TEM image, it is clear to see both “angular” and “flat” surface on JNJ-890 or CAM-A treated Cp149 HBV capsid. The angular morphology compared to flat morphology would already suggest the capsid subunits were under different contact environment and yet the authors observe single conformation of the capsid protein. Could the authors elucidate this in more details? Furthermore, if the angular morphology is under the detection limit of NMR, say 5%, it would suggest that less than 1 pentameric vertex ($12 * 5\% = 0.6$) appeared in every particle. Again, this is unlikely based on the observed TEM image. Finally, even at the flatter structure, like Y132A with CAM-A (e.g. PDB:5wre), each subunit in the crystallographic asymmetric unit could still exist different conformation (at least more than 10 residues are different).

2. Continue above. The authors prepared Cp149 / JNJ-890 assembly under 150 mM NaCl. The salt concentration was further diluted into 75 mM for NMR experiment. It is known that under low salt concentration, Cp149 capsid is less stable and could be disassembled. Similarly, low temperature could also disassemble the capsid/CAM-A complex into capsid protein dimer. Therefore, under the experimental conditions (low salt, 4 degree, 200,000g, overnight) that are prone to disassemble the complex into dimer, how could the author confirm the recorded NMR spectra were from sample under the complex form as seen in TEM and not from a different conformation?

3. Likewise, what are the NaCl concentrations for both Cp183 and P7-Cp183 during the protein purification and in the NMR data collection?

4. What is the oxidizing and reducing effects to the Cp183/JNJ-890 complex? The complex seems to form larger assemblies when kept for a long time (more oxidizing capsid protein) as well as when added DTT (more reducing environment).

5. In P7-Cp183/JNJ-890 sample, how could the author confirm that it has the same CAM occupancy as the other types of capsids? P7-Cp183 seems to have more distinct population of different types of particles in the TEM images.

6. Minor comments: Fig. S2, b), it should be CAM-E in the figure label.

We thank the referees for their very thorough reading of the manuscript, their most helpful and constructive evaluation, and their very interesting comments. Please find our answers below. The modifications in the manuscript and SI are highlighted in yellow.

REVIEWER COMMENTS

Reviewer #1 (Remarks to the Author):

NCOMMS-22-35774

The manuscript of Lecoq et al describes in details the effect of capsid assembly modulators (CAMs) on different forms of the HBV capsid protein. This is an excellent working showing the power of solid-state NMR (SSNMR) to reveal structural details on native systems without reverting to mutant or truncated protein systems. While the effect of CAMs has been studied before by X-ray and cryoEM, those studies required mutants, truncated proteins and conditions that alter the performance of CAMs. In this study, the authors show how full-length HBV capsids (cp183) resist capsid disassembly in the presence of CAM-A unlike cp149 or the presence of DTT present in cryoEM sample preparations. Moreover, large aberrant changes observe in EM images, reproduced by adding DTT, are shown to be associated only with a transformation of symmetry but not of capsid structure. Similarly deformation of the truncated forms do not alter the atom structure of the Cps. Another key finding is that unlike prior suggestions CAM-E do not promote the generation of empty capsids but either undergoes phosphorylation or packages non pgRNA molecules that are required for capsid stabilization. These, and others, are key finding unroute to viral drug design, and also point to some key aspects usually underlooked in such studies, and that refer to the validity of studying such systems not using the original native protein forms. Methodology is sound, results are well documented an explained, and conclusions are well supported by the data. All experimental work is well documented and can probably be reproduced given the correct expertise in sample preparation and NMR experiments and equipment. One “unresolved” issue is the fate of RNA amount upon addition of CAM-E. A solution for that is suggested in the comments below.

There are several comments that authors have to address prior to publication.

(1) Page 5 – please explain what is the spike part of cp, and its importance. It is later suggested that enhancement dynamics in this region is a dominant drug effect.

>> We are sorry for this shortcoming. We now explain in the introduction how the spike is formed (p2: “The central helices $\alpha 3$ and $\alpha 4$ from two Cp molecules associate into a four-helix bundle, forming a stable dimer; these associations form the spikes which are a prominent feature of the capsid shell.”), cite previous reports on its flexibility, and mention its functional importance, notably in nuclear export as suggested by recent work (p3: “Typically, the Cp spikes have recently been described to be involved in nuclear export of the encapsidated viral RNA, with the tip containing a sensor for secretion^{15,16}. While the flexibility of the spike tips has been repeatedly reported^{17–19}, its functional importance is not firmly settled, but might be important in adequately presenting the nuclear export signals, and also other envelopment-relevant motifs.”).

(2) A. CSP studies across the manuscript: How are ¹³C CSPs calculated given there are several carbons per amino acid. Only Ca? RMSD of all assigned atoms? Some normalization factor? What is actually used for the plots in Figure 2c for example?

>> This was indeed not clear. Only the largest CSP is the one which is taken into account, as reported in Fig S5a, where the CSP of each assigned spin is shown as a black line in the individual grey/orange/red bars. We now describe this in more detail in Figure S5a caption (p11: “(derived from the largest individual CSP observed for a given amino acid)”).

B. In the SI there are tables with CSP values per atom. There is no meaning for using four digits there. Please reduce the number to the correct accuracy of SSNMR measurements (one digit probably correct)

>> We agree with the referee and rounded the values.

C. The ¹⁵N CSPs are scaled by the gamma ratios. I don’t understand the rational. The chemical shift range of nitrogen is ~40 ppm (100-140). That of carbon for example is larger, and that of ¹H smaller. Previous studies in the field used values of 0.25, or even different values for different amino acids depending on the BMRB distributions of

chemical shifts. When using H-N CSPs, the contribution of ¹⁵N is scaled by 0.01, making them unimportant (my guess is the CSP for H alone, and CSP for both will be equivalent). Please elaborate / correct if needed.

>> Thank you for having pointed this out. To cite Mike P. Williamson, "... There is no agreed weighting, but the weighting is typically approximately 0.14...". (Williamson, M.P. *Modern Magnetic Resonance* (2018); and Williamson, Mike P. *Progr NMR Spectr.* (2013).) In Williamson's work, the factor α of 0.14 is then squared as well. After having had a closer look, we definitely agree with the referee that there is no relationship to the gyromagnetic ratios, and thus removed the term. Still, we kept the factors of 0.1 or 0.4, depending on whether the nitrogen CS were used together with ¹H or ¹³C, since we find that they account rather well for the chemical-shift range of a single kind of spin: H(N) (ca. 3 ppm), Ca (ca. 12 ppm), and N(H) (ca. 30 ppm). Since only ¹H and ¹⁵N were scaled, but not ¹³C, we did not have to worry about the more diverse chemical-shift range of the ¹³C side chain spins.

D. How do you calculate CSPs when you have split peaks? What is used?

>> Sorry, this was indeed not clear. In case of split peaks, the most intense peak was used to calculate CSPs, as now illustrated in Figure S1 and added in the caption p6: "For split peaks, the most intense signal was used for CSPs."

(3) In two cases the authors deduce that reduced line intensities are either from reduction in the amount of RNA or dynamics – from ¹³C DARR experiments and from ³¹P CP experiments.

A. Can the authors measure ¹³C INEPT spectra to assess if the RNA indeed becomes dynamic? At least some of the ribose peaks are unique and will not overlap in 1D INEPT with protein peaks.

>> 2D ¹H-¹³C INEPT show already in the absence of CAM many signals of nucleic acids. To evaluate whether the RNA becomes more flexible would require a quantitative analysis. Although peak groups can be formed, as we have shown previously (e.g., Callon et al., *Angew. Chem* 2022), we found that the 2D INEPT spectra of different CAM samples look too different to be useful for a detailed (quantitative) analysis without full resonance assignment, which is difficult to obtain.

B. Regarding the ³¹P spectra – the authors need a really large amount of scans to get a 1D signal (<40000) which I find strange. Did they try single-pulse excitation to obtain ³¹P signals? This way the spectrum will be quantitative and it will be possible to determine if the source of signal reduction is dynamics or reduced amount of RNA. With such simple experiments (¹³C INEPT, ³¹P 1D) a key question in the article will be answered explaining more about the mode of action of CAM-E.

>> The high number of scans is partially due to the low protein amount from the cell-free reaction pellets used to fill the 1.3 mm rotors. With already > 40 000 scans recorded for the CP, the much less sensitive direct pulse experiment would ask, for the four samples, several weeks of recording time, due to the much longer relaxation delay. We did thus not try single-pulse excitation.

(4) Another solution to obtain better RNA signals can probably be reduction in temperatures, again confirming the question of dynamics vs. amount of RNA.

>> Reduction of temperature could indeed be used, but would need recording of the already long spectra at different temperatures for all samples, which seems difficult to us.

(5) Page 20 – why would some CAM inhibit phosphorylation (e.g. HAP_R10) while others (JNJ-890) not in the WG media? Do they inhibit the kinases? This can be checked by production a different protein in their presence.

>> This is a good question which we have as of yet no answer to. That CAMs might inhibit kinases might be an interesting aspect to investigate.

Comments on SI:

(6) Preparation of ssNMR samples – cp183 with JNJ-632 and 890 where not cleared of Triton-X100. Is that because it is not there? It doesn't affect the spectra? Please explain. Can this have any impact on the results?

>> These two samples were not cleared of Triton-X100 because they have been prepared prior to the identification of the detergent in the capsid pocket (see Lecoq et al 2021). Triton-X100 affects the spectra in regions close to the Triton-X100 binding site (the Cp hydrophobic pocket), which is far from the CAM binding site (the Cp hydrophobic cavity). The presence of Triton-X100 does not impact results, as we show in Figure S2b.

(7) Figure S1 is the only one showing some peak splitting in the NH spectra for A137 and S35. The different peaks have completely different intensities. Why?

>> This is an interesting point which however we have no answer to yet. Since cross polarization efficiency is dependent on different parameters, like for instance local proton density and local dynamics, we just can conclude that one of these factors must be different in the four monomers and impact signal intensity.

(8) Figure S1 – can the H-N spectrum really resolve > 100 peaks as analyzed in (b)? It is difficult to see. Can you show one figure with 'x's marked on all the identified peaks?

>> The exact position of the peaks in the 2D H-N spectra was identified using 3D spectra (3D hCANH), where the resolution is sufficient to distinguish the 112 assigned residues. We added the cross-peaks in Figure S1 as requested by the reviewer, which were used for the CSP calculations, and added in the legend p6: "with cross-peaks corresponding to assigned residues". ... "Exact position of peaks in the 2D hNH were determined using 3D hCANH spectra."

(9) Figure S3: where are the results plotted on pdb 1qgt? There is one plot with the HAP18 but not the plot on free Cp149 ("left"). Also please fix "kJz".

>> Thank you for having spotted this. There was a mistake in the figure legend, the pdb 1qgt is actually shown in Figure 1g in the main text. We replaced the legend of Figure S3b on page 8 by: "Differences in $R_{1\rho}$ relaxation parameters mapped on the Cp149 structure (PDB 5d7y with HAP18 molecule shown in golden spheres)."

(10) Figure S5a – same question as above for ^{13}C CSPs. Please explain what we are seeing (I guess this is related to the black bars inside each colored bar, but this is not explained in the caption).

>> We added in the caption p11: "CSPs of the individual spins are shown as black lines inside the bars."

(11) Figure S15: second "green" is "pink"? What is considered large for HN CSP and what is small? (in numbers)

>> Thanks for catching this error, we corrected. Large would be > 0.2 ppm, small < 0.1 ppm, as now added in the caption p22: "..., which for the largest part are small (< 0.1 ppm), and none > 0.2 ppm."

(12) Table S2: missing "field" value

>> Thank you - the field was added in the Table p26.

(13) Tables S2: You seem to have performed DQ-DCP for N-C transfer at low spinning rate. This is very uncommon. Can you elaborate a bit? Give a reference? Show an example why this is better than SQ-DCP?

>> The DCP can either be DQ or ZQ (see e.g. Hediger S, Meier BH, Ernst R. 1995. Adiabatic passage ·Hartmann-Hahn cross polarization in NMR under magic angle sample spinning. *Chemical Physics Letters* **240**:449.). The two options are to first order equivalent in polarization-transfer efficiency. There is no theoretical reason why we choose one over the other. The theory in Hediger et al. is for a proton-carbon spin pair but is identical for carbon-nitrogen (Baldus M, Geurts DG, Hediger S, Meier BH. 1996. Efficient ^{15}N - ^{13}C polarization transfer by adiabatic-passage Hartmann-Hahn cross polarization. *Journal Of Magnetic Resonance Series A* **118**:140-144. In our hands, the two conditions yield the same amount of signal.

(14) Table S3: Please make sure fonts are consistent.

>> Thanks for seeing this. Done.

(15) Table S5: as written above – please correct significant figures.

>> The number of digits was reduced to 2.

Reviewer #2 (Remarks to the Author):

The manuscript "Molecular elucidation of drug-induced abnormal assemblies of the Hepatitis B Virus capsid protein by solid-state NMR" by Lauriane Lecoq addresses the off-path assembly of HBV nucleocapsid elicited by the Capsid Assembly Modulators (CAM). The mechanism by which CAM work is unknown and the authors used solid state NMR to get insights into this structural off-pathway capsid assembly. The manuscript is scientifically sounds and merits publication.

Major points

1. The effect of CAM (CAM-A and CAM-E) seems to show an interrelationship between anomalous assembly and the incorporation of nucleic acid. Both CAMs interact at the same site and led to the breakage of the quasi-equivalence in the asymmetric unit. In my point of view, it is this asymmetry that controls the correct assembly and at the same time the nucleic acid interaction. This manuscript showed unambiguously this effect, but it is lacking a discussion relating the relationship between symmetry in the nucleocapsid, quasi-equivalence in the asymmetric unit and molecular recognition of the virus genome.

>> This is a very interesting aspect. For non-specific packaging of RNA, the overwhelming factor is the basic CTD: Without CTD (Cp149) no significant RNA binding/packaging takes place in *E. coli*, and no pregenomic RNA encapsidation in HBV transfected/infected human cells. With unmodified CTD, in *E. coli* non-sequence-specific packaging (e.g. Porterfield et al. JVI 2010) of equivalent to about 4.000 nt of RNA is observed, but not when 7 of 8 hydroxy-amino-acids in the CTD are phosphorylated by SRPK1 - then no more RNA packaging is observed. On maturation, dynamic changes in CTD phosphorylation status accompany pgRNA packaging and reverse transcription. If capsid shells remain open due to CAMs, they cannot fully protect the nucleic acid; insofar it is indeed generally true that the "asymmetry controls correct assembly and ... nucleic acid interaction". However, we here actually can not comment on the molecular recognition of the virus genome - which specific recognition was indeed suggested to depend on the quasi-equivalence in the asymmetric unit (Patel et al. Nature Microbiology 2017) - since we do not use pgRNA, nor polymerase here. We however added a sentence and reference to this dependence in the discussion p20: "How the molecular recognition of the viral genome, which has been shown to be dependent on the quasi-equivalence in the asymmetric unit⁸⁰, would be impacted by CAMs remains to be established."

2. The authors state that "additional inter-dimer interactions in Cp183 which maintain a roughly spherical structure of the multimer". Although this is valid and even the most probable explanation, but one cannot exclude the possibility that the C-terminal IDR have an entropic contribution to the capsid assembly without any direct interaction. Especially when symmetry/asymmetry is involved in the assembly process. There are many studies correlating entropy and symmetry.

>> Indeed - we added a sentence and a reference referring to this possibility on p11: "... , even if one cannot exclude the possibility that the C-terminal intrinsically disordered regions have an entropic contribution to the capsid assembly without any direct interaction⁵⁸."

3. In the discussion session, the authors state that "single molecule is present in the asymmetric unit of the aberrant assemblies formed in the presence of CAM-A". Does the authors are considering the possibility of an aberrant particle with T=1. My feeling is that this would be unlikely. It would be more likely to have looser contacts induced by CAM-A, breaking the asymmetry in the asymmetric unit, what would make the icosahedral symmetry unlikely.

>> Thanks for this interesting consideration – no, we did not consider an aberrant particle with T=1. Besides that this has not been seen in EM, it also seems theoretically difficult, since the dimers need to be sitting across the 2-folds, and the hand-regions meeting at the 3-folds. Even if Cp can adapt to having 4 or 5 neighbors in T=3 and T=4 particles, we guess that having only 2 neighbors would likely not be a stable arrangement.

4. The effect of DTT on the Cp183 remained unclear. The authors made it clear by the mutant C183A that C183 is not involved in disulfide bond. However, it was not clear if DTT directly interacted with the core protein. Was there any CSP upon addition of DTT?

>> No, capsids in presence or absence of DTT yield the same spectra. We added this in Fig. S9 caption p16: "Capsids in absence of DTT yield same spectra, with the exception that they can reflect a different oxidation state of the cysteines."

5. In the experiments of cell free capsid assembly, it would be interesting to see the presence of the dimers or tetramers of Cp183 in solution. Studies have shown that the assembly process may obey the law of mass action of thermodynamics, contributing to the total entropic process (2nd law). Do the authors have any measurements of the relative concentration of dimers, tetramers and capsids in the experiments described in Figure 4 and S14?

>> We have not observed the presence of dimers or tetramers in the CF studies of Cp183; however, Cp149 does not assemble at all in the CF reaction (Wang et al., Frontiers Mol. Biosc. 2019), and these entities must be present there. Still, we do not have any quantitative measure for either.

6. There are no R1rho information (gaps) for the residues near the described CAM binding site. They are lacking assignment or are in conformational exchange?

>> While the CAM bound form is mostly characterized, many apo corresponding Cp149 signals are missing from the spectrum due to limited signal-to-noise ratios, which are divided by the peak multiplicity, and some few show overlap. In addition, all proline residues (there are 4 in the CAM binding site) are not observed.

Minor points

1. Figure S3 – While it is clear in the text, the legend does not state the difference between free and bound.

>> Thanks for catching this. As noticed by reviewer #1 also, we replaced the legend of Figure S3b on p8: “Differences in $R_{1\rho}$ relaxation parameters mapped on the Cp149 structure (PDB 5d7y¹⁷ with HAP18 molecule shown in golden spheres).”

2. It would be helpful to the reader to have a Supplementary table with CAM-E and CAM-A compounds

>> Good point. We provided this now in new Table S1 on page 24 with the detailed formula and molecular weights.

Reviewer #3 (Remarks to the Author):

The manuscript entitled “Molecular elucidation of drug-induced abnormal assemblies of the Hepatitis B Virus capsid protein by solid-state NMR” by Lecoq et al reports using solid-state NMR to study various E Coli expressed, pre-assembled HBV capsids interacted with different types of capsid assembly modulators (CAM). They found that CAM-A (formation of aberrant assemblies) induced aberrant structures, despite in an opened conformation, present one single conformation of the capsid protein. This is different to the apo HBV capsids that contain four different conformations of the capsid protein subunit in the icosahedral contact. Furthermore, the authors found the assembly domain of the full-length Cp183 capsid (either unphosphorylated or fully phosphorylated) shows the same conformation at the molecular level to the C-terminal truncated Cp149 capsid when treated with CAM-A despite they looked quite different under the negative stain TEM. Finally, the author examined the effect of CAM to the capsid protein subunit during the assembly process using cell-free translation system.

Overall, the manuscript is very interesting and it represents an important addition to the field. It is of general interest for the readership of Nature Communication. However, I also found some results would need to be further clarified before the manuscript can be accepted for publication.

1. Based on the negative stain TEM image, it is clear to see both “angular” and “flat” surface on JNJ-890 or CAM-A treated Cp149 HBV capsid. The angular morphology compared to flat morphology would already suggest the capsid subunits were under different contact environment and yet the authors observe single conformation of the capsid protein. Could the authors elucidate this in more details?

>> We have tried to present the different scenarios in the discussion starting on p. 17. We could additionally mention protein fibrils, which often show a large heterogeneity under the microscope. Nevertheless, we observed that regardless of this, the NMR peaks range from very narrow to very broad for different fibrils, reflecting the degree of local order, but showing no correlation with the heterogeneity in the micrographs. We also observed that dengue capsid ribonucleoprotein yields narrow NMR lines for objects that look like heterogeneous aggregates under the microscope. However, we have refrained from discussing these other systems as we believe this would be going too far. At present, what is well established is that multiple molecules in an asymmetric unit often give rise to multiple peaks that can be resolved if the arrangement leads to sufficiently different conformations. How smaller inhomogeneities are reflected in the spectra is less clear, but if one had an infinitesimally small linewidth, one could probably resolve conformations that were only slightly different. We added in the Discussion: “Nevertheless, minute heterogeneities could only be observed if the NMR lines tended toward infinitesimally small,” which, unfortunately, is not the case today.

Furthermore, if the angular morphology is under the detection limit of NMR, say 5%, it would suggest that less than 1 pentameric vertex ($12 * 5\% = 0.6$) appeared in every particle. Again, this is unlikely based on the observed TEM image.

>> Thank you. We added this calculus, which makes a clearer point than what we had stated before, to the discussion p18: “the residual presence of pentameric vertices which remain below the detection limit at the current signal-to-noise ratio, i.e. < 5 %. This would however suggest that less than one pentameric vertex ($12 * 5\% = 0.6$) appeared in every particle. This seems however unlikely based on the observed micrographs.”

Finally, even at the flatter structure, like Y132A with CAM-A (e.g. PDB:5wre), each subunit in the crystallographic asymmetric unit could still exist in a different conformation (at least more than 10 residues are different).

>>An overlay of the three dimers in the asymmetric unit of PDB 5e0i (Klumpp et al., PNAS 2015) shows that the structures vary only in the stalk regions. These are however involved in crystal contacts; it is thus unclear whether the dimers would be different in a flat arrangement outside the context of the crystal. NMR of the crystals might give a partial answer with respect to the equivalence by a possible peak multiplicity, but we did not consider this experiment, also as the results might not add enough insight from a biological perspective.

2. Continue above. The authors prepared Cp149 / JNJ-890 assembly under 150 mM NaCl. The salt concentration was further diluted into 75 mM for NMR experiment. It is known that under low salt concentration, Cp149 capsid is less stable and could be disassembled. Similarly, low temperature could also disassemble the capsid/CAM-A complex into capsid protein dimer. Therefore, under the experimental conditions (low salt, 4 degree, 200,000g, overnight) that are prone to disassemble the complex into dimer, how could the author confirm the recorded NMR spectra were from sample under the complex form as seen in TEM and not from a different conformation?

>> This is an interesting remark. In order to confirm that experimental conditions used during NMR sample preparation did not induce disassembly of the complex into dimers, we now analyzed representative resuspension of left-overs from sediments from NMR rotor filling and also a sediment removed from the rotor after the NMR experiment by negative staining EM, as now shown in Figure S4b. The micrographs reveal similar capsids or opened objects as the pictures taken from the samples before rotor filling and centrifugation. We added a Figure S4b and caption on p10: “In order to confirm that experimental conditions used during NMR sample preparation (low salt, 4 °C, ultracentrifugation) do not induce disassembly of the complexes into dimers, we analyzed representative resuspension of left-overs from sediments from NMR rotor filling and also a sediment removed from the rotor after the NMR experiment by negative staining EM. a) ^{13}C - ^{15}N -Cp149+JNJ-632; b) ^{13}C - ^{15}N -Cp149+JNJ-827; c) ^{13}C - ^{15}N -Cp149+JNJ-890; d) ^2H - ^{13}C - ^{15}N -Cp149 +JNJ-890 (from NMR rotor). The micrographs reveal similar capsids or opened objects as the pictures taken from the samples before rotor filling and centrifugation. “

3. Likewise, what are the NaCl concentrations for both Cp183 and P7-Cp183 during the protein purification and in the NMR data collection?

>> There was actually no NaCl during Cp183 and P7-Cp183 purification and NMR data collection. We added the buffer description in the SI Material and Methods p2: “For each sample, between 15 and 20 mg of autoassembled capsids at a concentration ~1-1.5 mg/ml were incubated at a monomer:CAM ratio of 1:4 for 2 h at 37 °C in 50 mM TRIS buffer at pH 7.5 in presence of 5 mM DTT.”

4. What are the oxidizing and reducing effects to the Cp183/JNJ-890 complex? The complex seems to form larger assemblies when kept for a long time (more oxidizing capsid protein) as well as when added DTT (more reducing environment).

>> This is an interesting issue, for which we do not have an explanation yet - we added a sentence, and also a reference to previous observations related to oxidation, for further discussion, on p19: “While the oxidation state, notably of C61, has shown to impact capsid stability⁷⁷, it remains an open question today how it can induce at the molecular level the more readily observed capsid opening in presence of CAM-A both under reducing (DTT) or oxidizing (longer time delays) conditions.”

5. In P7-Cp183/JNJ-890 sample, how could the author confirm that it has the same CAM occupancy as the other types of capsids? P7-Cp183 seems to have more distinct population of different types of particles in the TEM images.

>> This is given by the observation of a single signal, like in other capsids in presence of CAM-A. If different occupancies would be present, we would observe additional peaks corresponding to a free site.

6. Minor comments: Fig. S2, b), it should be CAM-E in the figure label.

>> Thank you very much for this, we changed.

Reviewers' Comments:

Reviewer #1:

Remarks to the Author:

I have gone through the revisions carefully and all comments have been answered satisfactory while I am convinced that some proposed experiments are just not possible or are out of the scope of the current manuscript. The paper is therefore can be published in its current form.

Reviewer #2:

Remarks to the Author:

The revised version of the manuscript entitled "Molecular elucidation of drug-induced abnormal assemblies of the Hepatitis B Virus capsid protein by solid-state NMR", authored by Anja Böckmann and cols., answered all the concerns raised in first version. The manuscript is an important contribution to the understanding capsid formation and the effect of CAMs. It shows the importance of solid-state NMR to direct observe the quasi-symmetry. At this point I recommend the full acceptance of the manuscript.

Reviewer #3:

Remarks to the Author:

In the revised manuscript, Lecoq and colleagues have addressed and clarified all technical questions raised in a satisfactory manner. My main concern with the manuscript was the claim made regarding a single molecule or conformation presented in the asymmetric unit of Cp149 with CAM-A. If there was only one conformation in the asymmetric unit, the assembly would either be in a form of free dimer or a hexagonal sheet-like structure. The authors excluded the possibility of dimer conformation with new data added in Fig. S4b; therefore, leaving the sheet-like structures to be the likely result. It is possible that multiple sheet-like structures were formed simultaneously as shown in Fig. S4b-b). However, the sample after centrifugation (Fig. S4b-d) showed qualitatively smaller fragments than the sample before rotor filling (Fig. S4b-c), indicating that the sheet-like structure is quite fragile. Therefore, it is still unclear to me if the authors recorded the NMR spectra on the "sheet-like structure" or the free "trimer of dimers", but I assume that the spectra would be quite similar or the subtle difference is beyond the limit of detection. Finally, the authors suggested that Cp183 WT and P7-Cp183 capsids treated with CAM-A had the similar molecular interaction to Cp149 capsid treated with CAM-A; an implication of flat or planar conformation. It would be great if the authors can add a supplementary figure for zoom-in spectra showing Y132 for Cp149 Apo, Cp183 Apo, P7-Cp183 Apo, Cp149 + JNJ-890, Cp183 + JNJ-890, Cp149 + HAP_R10, and P7-Cp183 + HAP_R10. I would assume capsids treated with CAM-A would have one distinct spectrum at Y132 than Apo capsids.

We thank the referees for their positive remarks.

Reviewer #1 (Remarks to the Author):

I have gone through the revisions carefully and all comments have been answered satisfactory while I am convinced that some proposed experiments are just not possible or are out of the scope of the current manuscript. The paper is therefore can be published in its current form.

Reviewer #2 (Remarks to the Author):

The revised version of the manuscript entitled "Molecular elucidation of drug-induced abnormal assemblies of the Hepatitis B Virus capsid protein by solid-state NMR", authored by Anja Böckmann and cols., answered all the concerns raised in first version. The manuscript is an important contribution to the understanding capsid formation and the effect of CAMs. It shows the importance of solid-state NMR to direct observe the quasi-symmetry. At this point I recommend the full acceptance of the manuscript.

Reviewer #3 (Remarks to the Author):

In the revised manuscript, Lecoq and colleagues have addressed and clarified all technical questions raised in a satisfactory manner. My main concern with the manuscript was the claim made regarding a single molecule or conformation presented in the asymmetric unit of Cp149 with CAM-A. If there was only one conformation in the asymmetric unit, the assembly would either be in a form of free dimer or a hexagonal sheet-like structure. The authors excluded the possibility of dimer conformation with new data added in Fig. S4b; therefore, leaving the sheet-like structures to be the likely result. It is possible that multiple sheet-like structures were formed simultaneously as shown in Fig. S4b-b). However, the sample after centrifugation (Fig. S4b-d) showed qualitatively smaller fragments than the sample before rotor filling (Fig. S4b-c), indicating that the sheet-like structure is quite fragile. Therefore, it is still unclear to me if the authors recorded the NMR spectra on the "sheet-like structure" or the free "trimer of dimers", but I assume that the spectra would be quite similar or the subtle difference is beyond the limit of detection. Finally, the authors suggested that Cp183 WT and P7-Cp183 capsids treated with CAM-A had the similar molecular interaction to Cp149 capsid treated with CAM-A; an implication of flat or planar conformation. It would be great if the authors can add a supplementary figure for zoom-in spectra showing Y132 for Cp149 Apo, Cp183 Apo, P7-Cp183 Apo, Cp149 + JNJ-890, Cp183 + JNJ-890, Cp149 + HAP_R10, and P7-Cp183 + HAP_R10. I would assume capsids treated with CAM-A would have one distinct spectrum at Y132 than Apo capsids.

-> We have created a plot of the spectra as suggested which is shown in Supplementary Figure 5c, and which indeed shows the green distinct peak for Y132 in presence of CAM-A.